# TGM6 is a helminth secretory product that mimics TGF-β binding to TGFBR2 to antagonize signaling in fibroblasts

Stephen E. White[1,6], Tristin A. Schwartze[1], Ananya Mukundan [1], Christina Schoenherr [2], Shashi P. Singh [3,7], Maarten van Dinther[4], Kyle T. Cunningham[3], Madeleine P. J. White[3], Tiffany Campion [3], John Pritchard [2], Cynthia S. Hinck[1], Peter ten Dijke [4,8], Gareth J. Inman [2,5,8], Rick M. Maizels[3,8] & Andrew P. Hinck [1,8] ✉

TGM6 is a natural antagonist of mammalian TGF-β signaling produced by the murine helminth parasite *Heligmosomoides polygyrus*. It differs from the previously described agonist, TGM1 (TGF-β Mimic-1), in that it lacks domains 1/2 that bind TGFBR1. It nonetheless retains TGFBR2 binding through domain 3 and potently inhibits TGF-β signaling in fibroblasts and epithelial cells, but does not inhibit TGF-β signaling in T cells, consistent with divergent domains 4/5 and an altered co-receptor binding preference. The crystal structure of TGM6 bound to TGFBR2 reveals an interface remarkably similar to that of TGF-β with TGFBR2. Thus, TGM6 has adapted its structure to mimic TGF-β, while engaging a distinct co-receptor to direct antagonism to fibroblasts and epithelial cells. The co-expression of TGM6, along with immunosuppressive TGMs that activate the TGF-β pathway, may minimize fibrotic damage to the host as the parasite progresses through its life cycle from the intestinal lumen to submucosa and back again. The co-receptor-dependent targeting of TGFBR2 by the parasite provides a template for the development of therapies for targeting the cancer- and fibrosis-promoting activities of the TGF-βs in humans.

Helminths, which have co-evolved with their mammalian hosts over long evolutionary timescales, persist by secreting soluble factors that suppress key immune signaling pathways and modulate host immunity[1–5]. In recent studies, we showed that upon infection, the murine intestinal helminth *Heligmosomoides polygyrus* (*H. polygyrus*) secretes a protein known as TGF-β mimic, or TGM, that binds directly to the host receptors to activate the TGF-β pathway[6]. Thus, like the native cytokine, this stimulates the expression of the key

transcriptional regulator Foxp3 in naïve T cells, expanding the population of CD4[+] CD25[+] Foxp3[+] regulatory T cells (Tregs)[7,8]. The increased numbers of Tregs promote peripheral immune tolerance and are required for the persistence of *H. polygyrus* in its mammalian host[2,9–11].

The three mammalian TGF-β isoforms, TGF-β1, -β2, and -β3, control and influence many pathways in cellular differentiation[12–14] and are required for mediating immune tolerance[8,12,15] and maintaining the

[1]Department of Structural Biology, University of Pittsburgh School of Medicine, Pittsburgh, PA, USA. [2]Cancer Research UK Scotland Institute, University of Glasgow, Glasgow, UK. [3]Centre for Parasitology, School of Infection and Immunity, University of Glasgow, Glasgow, UK. [4]Oncode Institute and Department of Cell and Chemical Biology, University of Leiden, Leiden, The Netherlands. [5]School of Cancer Sciences, University of Glasgow, Glasgow, UK. [6]Present address: Ten63 Therapeutics, Durham, NC, USA. [7]Present address: Department of Biological Sciences, Birla Institute of Technology and Science-Pilani, Pilani, Rajasthan, India. [8]These authors jointly supervised this work: Peter ten Dijke, Gareth J. Inman, Rick M. Maizels, Andrew P. Hinck. ✉e-mail: ahinck@pitt.edu

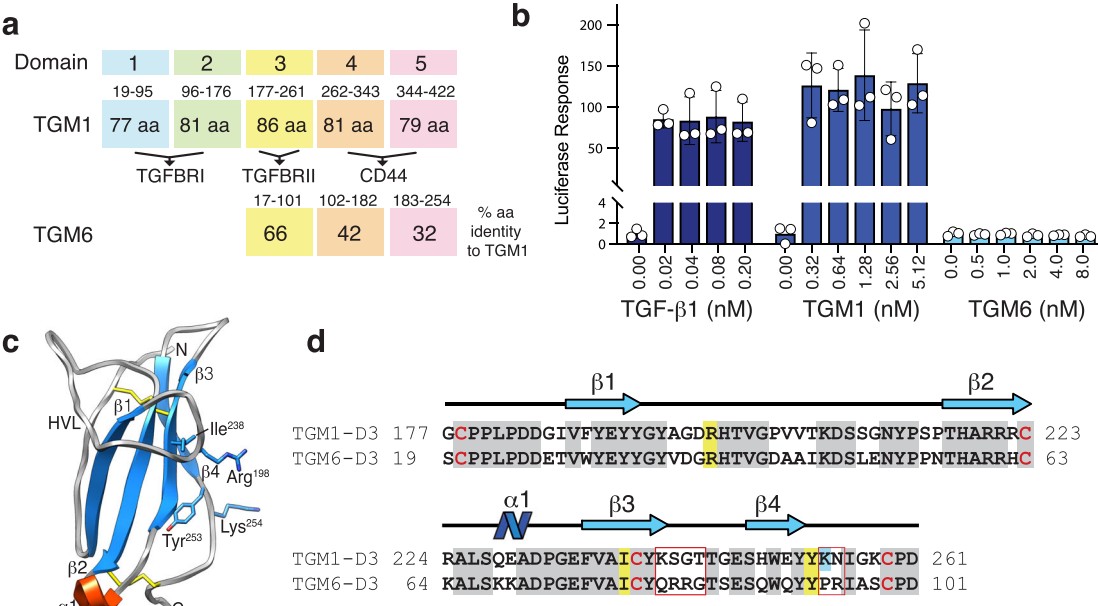

**Fig. 1 | TGM6 domain structure, similarity to TGM1, and signaling activity.**
**a** Comparison of the domain structure and amino sequence identity of TGM6 relative to TGM1. Residue numbering is shown above the shaded boxes corresponding to each domain. **b** TGF-β1-, TGM1-, and TGM6-simulated luciferase reporter activity in NIH-3T3 fibroblasts. Data shown are the mean and standard deviation of triplicate measurements from one of two experiments with similar results. **c** Structure of TGM1-D3 (PDB 7SXB)[27]. The sidechains of Arg[198], Ile[238], Tyr[253], and Lys[254] shown to be critical for binding are displayed in blue. **d** Sequence alignment of TGM1-D3 and TGM6-D3. Residues shown to be essential in TGM1-D3

for binding TGFBR2 and are conserved in TGM6 are shaded yellow; essential residue that is non-conserved in TGM6 is shaded cyan; all other residues that are conserved in TGM1 and TGM6 are shaded gray (except for conserved cysteines, which are shaded red). Residues highlighted by the red box are proposed to underlie the differential affinity of TGM6-D3 and TGM1-D3 for TGFBR2 (see "Results" section and Fig. 8). Note, the residues numbers of TGM1-D3 and TGM6-D3 differ by 160 due the presence of D12 only in TGM1. Source data of (**b**) provided as a Source Data file.

---

expression of proteins of the extracellular matrix, such as type I collagen and fibronectin[16]. The knockout of endogenous TGF-β1 in mice is characterized by the development of multi-organ inflammatory disease and death after maternal TGF-β1 is depleted[12]. The dysregulation of TGF-β signaling has also been shown to drive the pathogenesis of several human diseases, including inflammatory bowel disease[17], cancer[18,19], and renal, pulmonary, and cardiac fibrosis[16,20].

TGF-β growth factors are comprised of two elongated cystine-knotted monomers held together by a single interchain disulfide bond[21]. The growth factors signal by assembling a heterotetrameric complex with two near autonomously signaling pairs of serine/threonine kinase receptors, known as the TGF-β type I and type II receptors, TGFBR1 and TGFBR2[22–25]. This triggers a phosphorylation cascade, with constitutively active TGFBR2 phosphorylating TGFBR1, and activated TGFBR1 phosphorylating the downstream transcriptional effector molecules, SMAD2 and SMAD3[26].

In contrast to mammalian TGF-β, the helminth TGM molecule is a disulfide-rich 422 amino acid protein with an N-terminal signal peptide followed by five homologous domains[6]. These domains bear no homology to TGF-β or other TGF-β family members; instead, the individual domains are distantly related to the complement control protein (CCP) or sushi domain family[6,27]. There are at least nine homologs of TGM in *H. polygyrus*, which are numbered TGM2 through TGM10, with the founding member, TGM, being numbered TGM1[28]. Among this family of proteins, six (TGM1–6) are expressed primarily in adult parasites while the remaining four (TGM7–10) are expressed in the larvae[28,29]. Domains 1, 2, and 3 (D1, D2, and D3, respectively) of TGM1 are necessary and sufficient for SMAD-dependent signaling in reporter cells[28], with D1–D2 (D12) binding TGFBR1 with a $K_D$ of 30 nM and D3 binding TGFBR2 with a $K_D$ of 1.2 µM[27] (Fig. 1a). Domain 4 (D4), together with domain 5 (D5) (D45) bind CD44, a cell surface receptor that is abundant on T cells, which both targets and potentiates cellular

responsiveness to TGM1[30] (Fig. 1a). Among the TGMs expressed during the adult stages of the parasite, TGM6 is unique in that it lacks D1 and D2 (Fig. 1a) and it does not activate TGF-β signaling, unlike TGM1, TGM2, and TGM3 which signal in both fibroblasts and T cells, or TGM4 which signals in T cells and myeloid cells[6,28,31].

In this work, we show that TGM6, whose D3 shares 66% identity to TGM1-D3 (Fig. 1a), binds TGFBR2, but does not bind TGFBR1 or other type I and type II receptors of the TGF-β family. In TGF-β reporter assays in fibroblasts and epithelial cells, TGM6 potently inhibits TGF-β- and TGM1-induced signaling, consistent with its receptor binding profile, including its ability to compete with TGF-β for binding TGFBR2. Domains 4 and 5 (D45) of TGM6 are, however, divergent from D45 of TGM1, with only 42% and 32% identity (Fig. 1a). In accord with this, TGM6 does not bind CD44, and unlike TGM1 which is targeted to and active on T cells due to its ability to bind CD44 through D45, TGM6 is not targeted and is inactive on T cells. In addition, we present the crystal structure of the TGM6-D3:TGFBR2 complex, in which TGM6-D3 is shown to fully mimic binding of mammalian TGF-β to TGFBR2, presenting a similar convex surface with a hydrophobic interior and charged residues on the periphery and engaging the same set of residues. Together, these results suggest that TGM6 has adapted its domain structure and sequence to mimic binding of mammalian TGF-β to TGFBR2 and to antagonize TGF-β and TGM signaling in fibroblasts— but to do so without interfering with essential immune-suppressive signaling of TGM agonists in T cells. The co-expression of the TGF-β antagonist TGM6, along with TGM agonists such as TGM1-4 that suppress immunity, may minimize fibrotic damage to the host as the parasite progresses through its life cycle from the intestinal lumen to submucosa and back again. The co-receptor-dependent targeting of TGF-β agonists and antagonists by the parasite provides a template for the development of therapies for targeting the cancer and fibrosis-promoting activities of the TGF-βs in humans.

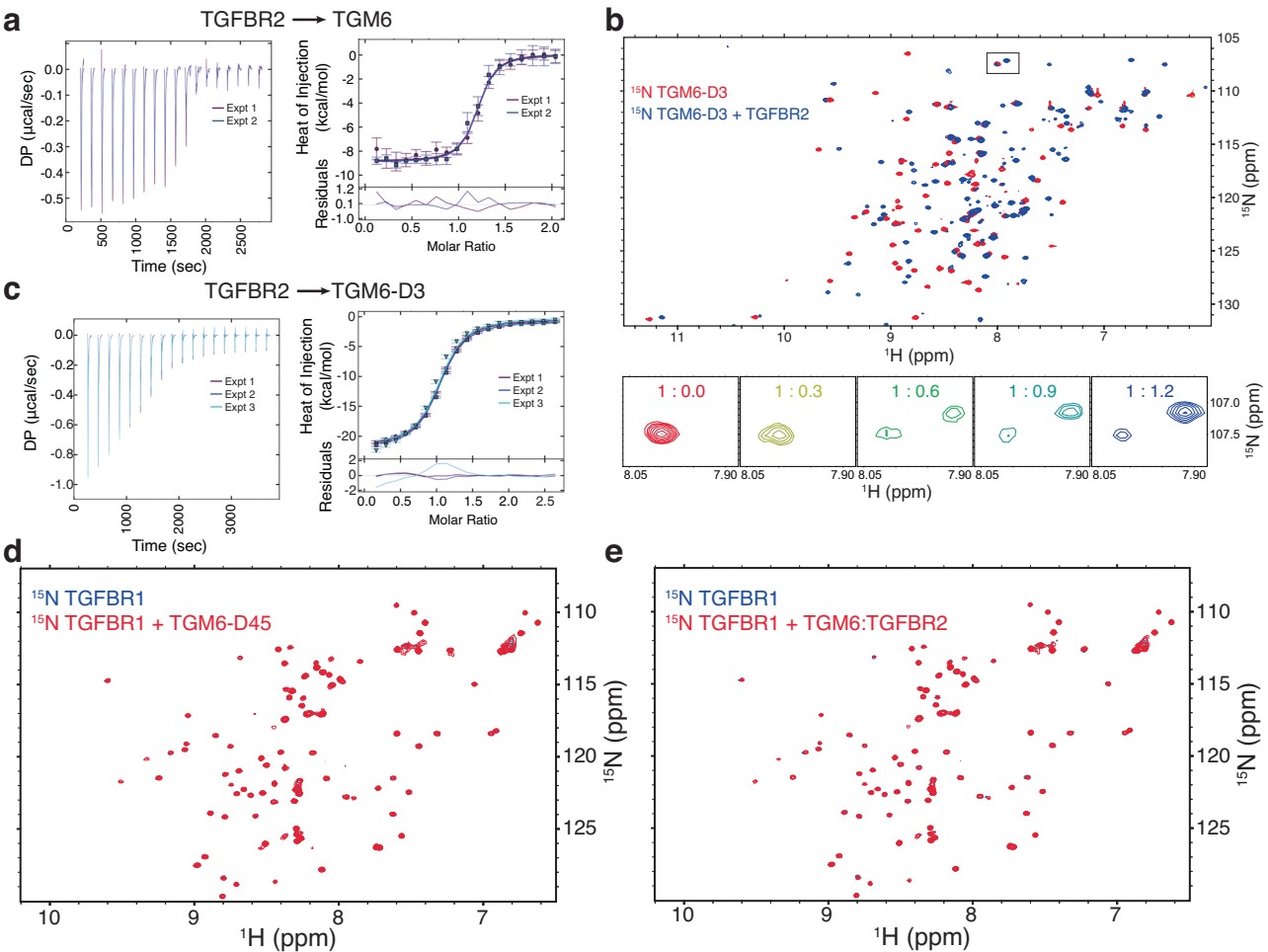

Fig. 2 | TGM6 binds TGFBR2 through D3. ITC thermograms (left) obtained upon injection of TGFBR2 into TGM6 (a) or TGM6-D3 (c). Thermograms are overlaid as two (a) or three (c) experiments shaded purple and blue and purple, blue, and cyan, respectively. Mean integrated heats and standard deviation (top) and accompanying fit to a 1:1 binding model are shown to the right of the thermograms with the residuals (bottom) as a function of the molar ratio. The difference between the $K_D$ for TGM6 and TGM6-D3 for binding TGBR2 is not statistically significant (two-sided unpaired $t$-test $p$ value = 0.18; assuming $n = 2$ for TGM6 and $n = 3$ for TGM6-D3 replicate experiment count). **b** $^1H–^{15}N$ HSQC spectra of $^{15}N$ TGM6-D3 alone (red) overlaid onto the spectrum of the same sample containing a 1.2-fold molar excess of unlabeled TGFBR2 (blue). Shown below is an expansion of the boxed region with all titration points labeled as the molar ratio of $^{15}N$ TGM6-D3:TGFBR2. **d, e** $^1H–^{15}N$ HSQC spectra of $^{15}N$ TGFBR1 alone (red) overlaid onto the spectrum of the same sample containing a 1.2-fold molar excess of unlabeled TGM6-D45 (blue) or TGM6:TGFBR2 complex (blue) (**d** and **e**, respectively). Source data of (**a**–**e**) provided through Figshare [https://doi.org/10.6084/m9.figshare.28179359].

## Results

### TGM6 lacks signaling activity, but selectively and specifically binds TGFBR2

In contrast to TGM1, TGM2, and TGM3, TGM6 was previously shown to lack detectable signaling activity with the MFB-F11 reporter cell line, which is based on mouse embryonic fibroblasts stably transfected with a SMAD3-sensitive CAGA reporter[32]. In consideration of the reported high specificity of the MFB-F11 reporter for signaling induced by TGF-βs, but not activins which also activate the SMAD2/3 branch of the TGF-β pathway, we used another murine reporter cell line, NIH-3T3 fibroblasts, also stably transfected with a CAGA promoter element, which are responsive to both TGF-βs and activins[33]. However, in accord with the previous MFB-F11 assay results, when tested at comparable concentrations, TGM6 was unable to activate the NIH-3T3 reporter above baseline, while both TGF-β1 and TGM1 robustly activated, even at the lowest concentrations tested (Fig. 1b).

In previous studies, we determined the structure of refolded bacterially expressed TGM1-D3 and showed that it adopted the overall fold of a CCP domain, but was expanded to open a potential interaction surface comprised of several residues near the C-terminus, as well as the tip of the long structurally ordered hypervariable loop (HVL),

which wraps around the domain and extends toward the C-terminus (Fig. 1c)[27]. Through NMR chemical shift perturbation mapping and site-directed mutagenesis, we showed TGM1-D3 contacts the same edge β-strand of TGFBR2 (β4) as TGF-β; moreover, we identified three residues near the C-terminus of TGM1-D3, Ile[238], Tyr[253], and Lys[254], and one residue on the tip of the HVL, Arg[198], which when mutated led to a >20-fold reduction in binding affinity for TGFBR2 (Fig. 1c). Three of these four residues are identical in TGM6-D3, and the fourth, Lys[254], has an arginine at an adjacent position, suggesting that TGM6 might also bind the edge β-strand of TGFBR2 through a similar surface (Fig. 1d).

Isothermal titration calorimetry (ITC) was used to determine if, in the absence of signaling activity, TGM6 might nonetheless bind TGFBR2. In titrations in which TGFBR2 was titrated into mammalian-produced TGM6, strong exothermic responses were observed, which after integration and fitting, yielded a $K_D$ of 220 ± 100 nM (Fig. 2a and Supplementary Table 1). To determine if TGM6-D3 holds the full capacity for binding TGFBR2, we produced $^{15}N$-TGM6-D3 in bacteria and showed using NMR that addition of TGFBR2 led to perturbations and slow-exchange binding of more than half the amide signals (Fig. 2b), indicative of specific high-affinity binding. In ITC measurements with TGM6-D3 and TGFBR2, strong binding was observed with a

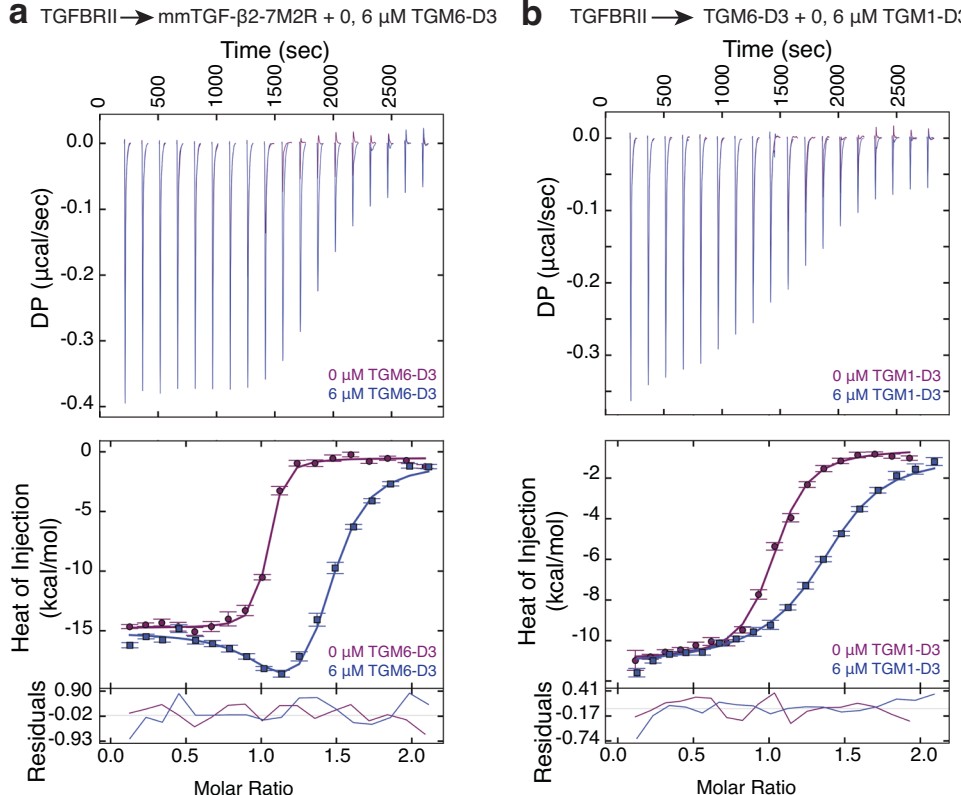

**Fig. 3 | TGM6-D3, and TGM1-D3, compete with TGF-β for binding TGFBR2.**
**a** Titration of TGFBR2 into mmTGF-β2-7M2R in the absence (purple) or presence (blue) of 6 µM TGM6-D3 as the lower-affinity binder. **b** Titration of TGFBR2 into TGM6-D3 in the absence (purple) or presence (blue) of 6 µM TGM1-D3 as the lower-affinity binder. Thermograms in (**a, b**) are presented as single experiment without

(purple) and with (blue) competitor. Each panel includes the thermograms (top), mean integrated heats ± SD and fitted isotherms (middle), and fitting residuals (bottom) for the associated titrations. The data were globally fit using a simple competitive binding model. Source data of (**a, b**) provided through Figshare [https://doi.org/10.6084/m9.figshare.28179359].

fitted $K_D$ of 440 ± 80 nM, which was not statistically different from the value obtained for full-length TGM6 (Fig. 2c and Supplementary Table 1). Thus, TGM6-D3 holds the full capacity for binding TGFBR2, yet unlike TGM1, the affinity of TGM6 for TGFBR2 is 4–5-times greater[27].

In the absence of TGM6 signaling activity in the MFB-F11 reporter line, we expected that TGM6 bound TGFBR1 weakly, or not at all. To test this, we recorded NMR spectra of [15N]-labeled TGFBR1 alone and with unlabeled TGM6-D45 added (Fig. 2d). The addition of unlabeled TGM6 D4-D5 (D45) led to no significant shifts in the signals of TGFBR1, even though the concentration was 100 µM and excess TGM6-D45 was added. This indicates that TGM6-D45 does not directly bind TGFBR1, even with moderate affinity. To test the possibility that binding of TGFBR1 is potentiated by TGFBR2, we re-recorded the spectra of [15N]-TGFBR1, but with addition of 1.1 equivalents of unlabeled TGM6:TGFBR2 complex, rather than TGM6-D45 alone (Fig. 2e). This also led to no significant shifts in the signals of TGFBR1, indicating that the TGM6:TGFBR2 complex also does not bind TGFBR1. The absence of shifts is not due to non-native folding of either TGFBR1 or TGM6-D45, which were produced in *Escherichia coli* (*E. coli*) and refolded, as the spectra of each are well-dispersed (Fig. 2d, e and Supplementary Fig. 1).

To test the possible binding of TGM6-D3 by other type II receptors of the TGF-β family, we used ITC to determine whether TGM6-D3 binds the activin and BMP type II receptors ActRII, ActRIIB, and BMPRII (Supplementary Fig. 2 and Supplementary Table 2). In this experiment, each type II receptor was titrated into TGM6-D3 or buffer alone, and in each case, there was no detectable binding. To test possible binding of TGM6 by other type I receptors of the TGF-β family, we prepared

[15N]-labeled BMP and activin type I receptors, ALK1, ALK2, ALK3, and ALK4, and recorded spectra with 1.1 equivalents of unlabeled TGM6-D45 (Supplementary Fig. 3) or TGM6:TGFBR2 complex (Supplementary Fig. 4) added, but like [15N]-TGFBR1, no shifts were observed. The native folding of the type I and type II receptors is demonstrated by the chemical shift dispersion of their 2D $^1H$–$^{15}N$ (Supplementary Figs. 3 and 4) or 1D $^1H$ spectra (Supplementary Fig. 5), ruling out the possibility that misfolding of these disulfide-rich receptors is responsible for the lack of binding. Thus, TGM6 also does not directly bind the BMP and activin type I and type II receptors, and in the case of type I receptors, binding is not potentiated by TGFBR2.

## TGM6-D3 competes for the TGF-β binding site on TGFBR2

TGM6-D3 binds TGFBR2 and the key residues for receptor binding by TGM1-D3 are conserved in TGM6-D3. Thus, it was hypothesized that TGM6-D3 would compete with TGF-β for binding TGFBR2, similar to TGM1-D3[27]. To test this, an ITC competition experiment was performed in which TGFBR2 was titrated into the engineered TGF-β monomer, mmTGF-β2-7M2R[34], either alone or in the presence 6 µM TGM6-D3 (Fig. 3a). The engineered TGF-β monomer was used for these experiments, rather than a native TGF-β dimer, due to its much higher solubility, yet unchanged TGFBR2 binding affinity compared to the native dimer. The addition of TGM6-D3 increased the extent of curvature in the binding isotherms and reduced the overall enthalpy of the reaction, consistent with the behavior expected for competitive binding as shown by the global fit of the data to a simple competitive binding model (Fig. 3a and Supplementary Table 3)[35]. Therefore, TGM6-D3 and TGF-β compete for the same binding site on TGFBR2. TGM6-D3 and TGM1-D3 are also expected to compete for binding

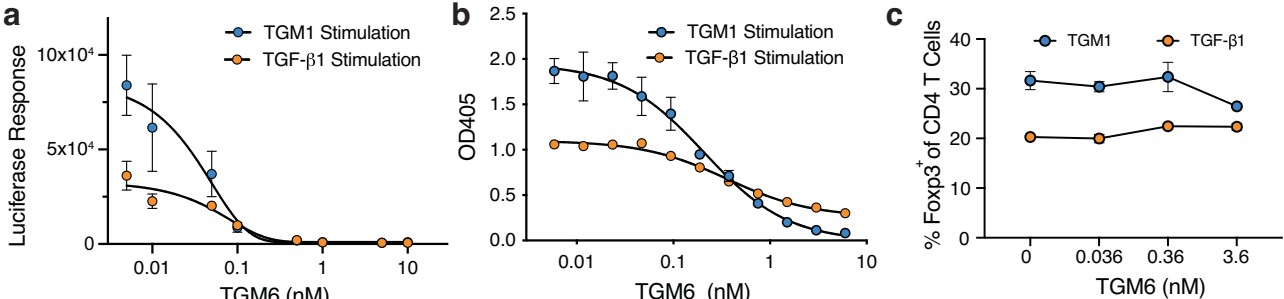

**Fig. 4 | TGM6 is a potent inhibitor of TGF-β and TGM1 signaling in fibroblasts, but not T cells.** Inhibition of SMAD2/3 CAGA reporter stimulated by TGF-β1 (orange symbols) or TGM1 (blue symbols) in NIH-3T3 (**a**) or MFB-F11 (**b**) fibroblasts by increasing concentrations of TGM6. Smooth black lines correspond to the fit of the data to a dose-dependent inhibition of TGF-β1 or TGM1 signaling by TGM6. **c** Inhibition of the TGF-β1 (orange symbols) or TGM1 (blue symbols)

induction of the Foxp3 transcription factor in murine splenic CD4+ T cells by increasing concentrations of TGM6. Data shown in (**a**, **b**) are mean ± standard deviation of triplicate measurements from one of two experiments with similar results. Data shown in (**c**) is the mean and standard deviation of triplicate measurements from one experiment. Source data of (**a**–**c**) provided as a Source Data file.

TGFBR2 based on their competitive binding with TGF-β. This was confirmed by performing a similar competition binding experiment, with TGFBR2 titrated into TGM6-D3 either alone or in the presence of 6 μM TGM1-D3 (Fig. 3b and Supplementary Table 3).

## TGM6 antagonizes TGF-β signaling in fibroblasts and epithelial cells, but not T cells

TGM6 binds TGFBR2, but does not bind TGFBR1, or any of the other BMP and activin type I and type II receptors tested, thus TGM6 might function as a TGF-β or TGM1 antagonist by occupying cell surface TGFBR2. To test this, NIH-3T3 fibroblast reporter cells were incubated with increasing concentrations of TGM6 prior to stimulation with TGF-β1 or TGM1 (Fig. 4a). The addition of TGM6 led to a dose-dependent decrease in signaling, with an $IC_{50}$ of ~0.05 nM for inhibition of both TGF-β1 and TGM1. The assay was repeated using the MFB-F11 reporter fibroblasts and TGM6 similarly led to a dose-dependent decrease in signaling, with an $IC_{50}$ of ~0.2 nM for inhibition of both TGF-β1 and TGM1 (Fig. 4b). However, when its ability to antagonize the conversion of murine splenic T cells to Foxp3+ Tregs by either TGF-β1 or TGM1 was measured, no inhibition was observed, even at concentrations that nearly fully inhibited TGF-β1 or TGM1 signaling in the NIH-3T3 or MFB-F11 reporter cells (Fig. 4c). To further investigate the range of cells in which TGM6 is active, TGM6 was tested for inhibition of TGF-β signaling in two additional mouse cell lines, EL4, a type of T-cell and NM18, a subclone of NMuMG breast epithelial cells[36]. TGM6 did inhibit signaling in NM18 epithelial cells, but it did not inhibit signaling in EL4 cells (Supplementary Fig. 6).

To test if TGM6 could serve as either an agonist or antagonist of BMP signaling, we used NIH-3T3 fibroblasts stably transfected with a BMP responsive element (BRE) coupled to a fluorescent (mCherry) reporter and treated the cells with either BMP2, BMP6, or BMP7 alone or with 3.56 nM TGM6 added. The BMPs stimulated the reporter, but TGM6 could neither stimulate the reporter nor inhibit reporter activity stimulated by the BMPs (Supplementary Fig. 7a). We also tested TGM6 for its ability to inhibit activin signaling using the NIH-3T3 fibroblasts stably transfected with a SMAD3 CAGA reporter element coupled to a fluorescent (GFP) reporter but observed that TGM6 was incapable of inhibiting activation of the reporter by activin A (ActA) (Supplementary Fig. 7b).

The finding that TGM6 inhibits TGF-β1 and TGM1 signaling in fibroblasts and epithelial cells, but not splenic or EL4 T cells, and that its inhibitory concentration in fibroblasts and epithelial cells is several thousand-fold lower than its affinity for TGFBR2 (ca. 0.1–0.4 nM vs. ca. 320 nM), suggests that its activity may be enhanced by a co-receptor, expressed by fibroblasts and epithelial cells but not T cells, that is specifically recognized and bound by TGM6-D45. To investigate this,

the NIH-3T3 and MFB-F11 reporter assays were repeated but using TGM6-D3 for inhibition instead of full-length TGM6 (Fig. 5a, b). This resulted in no inhibition below 1000 nM in either cell line or only minor inhibition in the MFB-F11 cell line at even higher concentrations, suggesting that D45 plays a critical role in the inhibition. To determine if physical attachment of D3 to D45 was required for inhibition, we compared treatment with D3, D45, or the combination of D3 and D45 (Fig. 5c). However, in contrast to the full-length protein at a concentration of 3.6 nM which completely inhibited signaling induced by TGF-β, there was no inhibition by the individual domains, or the combination at the same concentrations, suggesting that physical attachment of the domains is required for inhibition.

The inability to TGM6 to inhibit signaling in splenic T cells and EL4 T cells, together with the finding that efficient conversion of naïve T cells to Foxp3+ Tregs by TGM1 requires CD44 co-receptor binding by D45[30], suggested that these domains of TGM6 may not bind CD44. To investigate this, we used ITC to measure the binding affinity of TGM6-D45, and as a control TGM1-D45, for mouse CD44 (mCD44) (Fig. 5d–f). The titration of mCD44 into TGM6-D3 led to an insignificant response that did not change over the course of the titration, while titration of mCD44 into TGM1-D3 led to a robust exothermic response, which upon integration, yielded a $K_D$ in close accord with that reported earlier[30] (Supplementary Table 4). Thus, inhibition by TGM6 depends on D45, which cannot bind CD44, offering a likely explanation for its inability to inhibit TGF-β1 and TGM1 signaling in T cells.

## TGM6-D3 mimics TGF-β binding to TGFBR2

To identify the underlying molecular basis by which TGM6-D3 binds and recognizes TGFBR2, we isolated the TGM6-D3:TGFBR2 complex using size exclusion chromatography, screened for diffracting crystals, and determined the structure to a resolution of 1.40 Å (Fig. 6a and Supplementary Table 5). We found one TGM6-D3:TGFBR2 complex in the crystallographic asymmetric unit and interpretable density for residues 46–153 of TGFBR2 and residues 16–65 and 71–102 of TGM6-D3. Overlay of the TGM6-D3-bound TGFBR2 structure with the previous crystal structure of unbound TGFBR2 (PDB 1M9Z)[37] revealed only minor differences, with a backbone root mean square deviation (RMSD) of 0.49 Å (Supplementary Fig. 8a). Overlay of the TGFBR2-bound TGM6-D3 structure with the lowest energy unbound TGM1-D3 structure determined by NMR (PDB 7SXB)[27] revealed close similarity, with a backbone RMSD of 1.25 Å (Supplementary Fig. 8b). Residues missing in the density for TGM6-D3, Lys[66]–Ala[70], correspond to the tips of the loops connecting β-strands 2–3, which were previously shown to be flexible in unbound TGM1-D3[27].

There is a single large interface between TGM6-D3 and TGFBR2 with an area of 661 Å² (Fig. 6a). TGM6-D3 engages TGFBR2 primarily

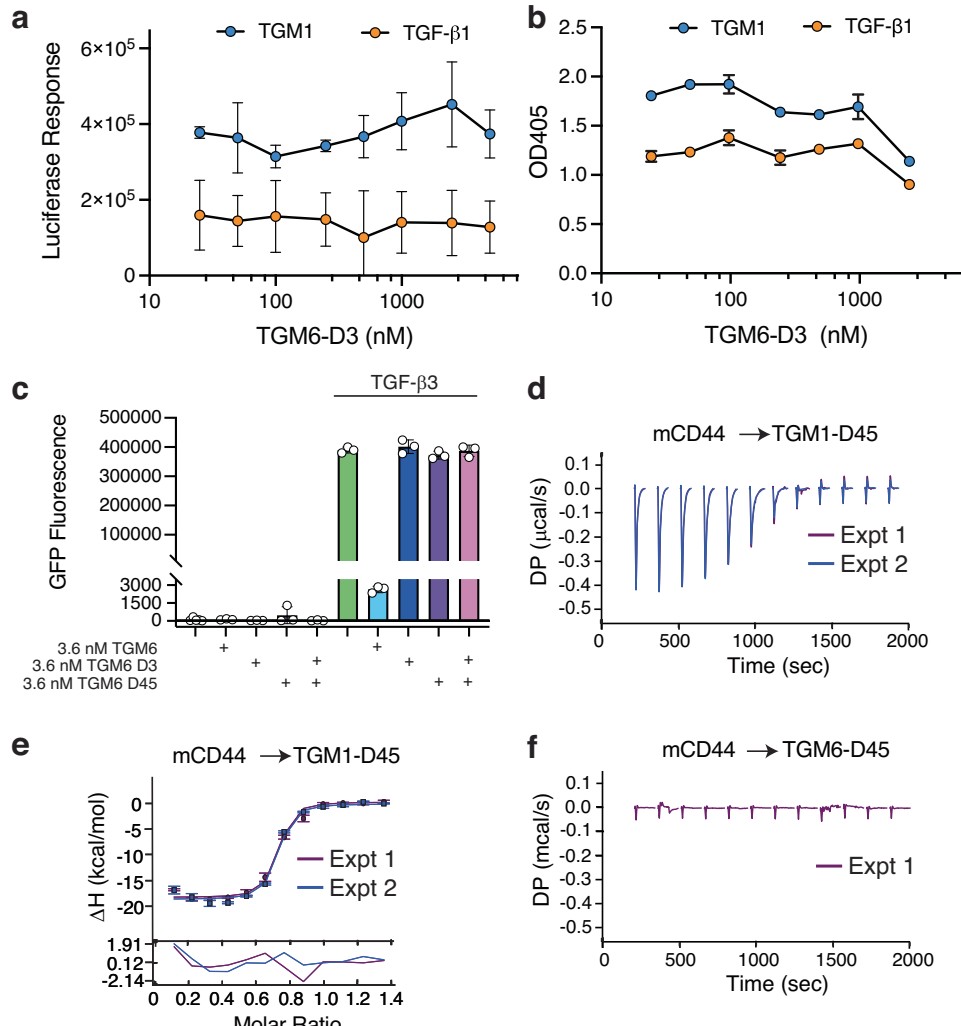

**Fig. 5 | TGM6 requires attachment of D3 to D45 to inhibit and does not bind the TGM1 co-receptor CD44.** Inhibition of TGF-β reporter stimulated by TGF-β1 (orange symbols) or TGM1 (blue symbols) in NIH-3T3 (**a**) or MFB-F11 (**b**) fibroblasts by increasing concentrations of TGM6-D3. Data could not be reliably fit to a dose-dependent inhibition model. **c** Inhibition of TGF-β CAGA-GFP reporter in NIH-3T3 fibroblasts by TGM6, TGM6-D3, TGM6-D45, or TGM6-D3 plus TGM6-D45. Data shown in (**a**) and (**b**) are mean and standard deviation of triplicate measurements of one experiment. Data shown in (**c**) are the mean and standard deviation of triplicate measurements from one of three experiments with similar results. ITC thermograms (**d**) and fitted isotherm (top) and residuals (bottom) (**e**) for two separate titrations of mCD44 into TGM1-D45. **f** ITC thermograms for a single titration of mCD44 into TGM6-D45. Source data of (**a**–**c**) and (**d**–**f**) provided as a Source Data file associated with the article or through Figshare [https://doi.org/10.6084/m9.figshare.28179359], respectively.

through the edge β-strand, β4, and does so through the interface near the C-terminus with the residues anticipated, Ile[78] from β-strand 3, Tyr[93] and Arg[95] from β-strand 4, and Arg[38] from the tip of the HVL[27] (Fig. 1c, d). The interface between TGM6-D3 and TGFBR2 is a remarkable mimic of the interface between TGF-β1/-β3 and TGFBR2[38,39], with a central hydrophobic region flanked by hydrogen-bonded ion-pairs at the periphery (Fig. 6). On one side of the interface, TGFBR2 Asp[55] and Glu[78] form hydrogen-bonded ion-pairs with TGM6-D3 Arg[95] and Arg[38], respectively (Figs. 6a and 7a), the first of which closely mimics the interaction between TGFBR2 Asp[55] and TGF-β1/-β3 Arg[394] (Figs. 6b and 7b). In the central hydrophobic region, TGFBR2 Ile[76] inserts into a hydrophobic pocket formed by TGM6-D3 residues Ile[78], Tyr[80], and Tyr[93] (Figs. 6a and 7c), mimicking the interaction between TGFBR2 Ile[76] and the hydrophobic pocket between the fingers of TGF-β formed by Trp[332], Tyr[390], and Val[392] (Figs. 6b and 7d). On the side of the interface opposite the TGM6-D3 Arg[95]:TGFBR2 Asp[55] interaction, TGFBR2 Asp[141] forms a hydrogen bond with the phenolic hydroxyl of TGM6-D3 Tyr[80] and TGFBR2 Glu[142] forms a hydrogen-bonded ion-pair with TGM6-D3 Arg[82]

(Figs. 6a and 7e), mimicking the hydrogen-bonded ion-pair between TGFBR2 Glu[142] and TGF-β1/-β3 Arg[325] (Figs. 6b and 7f).

### The interactions that enable TGM6-D3 and TGF-β1/-β3 to bind TGFBR2 are similar

Through mutagenesis and binding studies, the central hydrophobic interaction and two flanking hydrogen-bonded ion-pairs in the interface between TGF-β1/-β3 and TGFBR2 were each shown to contribute significantly to binding[40,41]. Though the overall architecture of the TGM6-D3:TGFBR2 interface is similar to the TGF-β1/-β3:TGFBR2 interface, it is about 50% larger (661 Å[2] vs. 479 Å[2] for the TGF-β1/-β3:TGFBR2 interface) and it has a greater number of hydrogen bond and ionic interactions (Fig. 6). Thus, to assess the contributions of these interactions in the TGM6-D3:TGFBR2 complex, we substituted single residues in both TGM6-D3 and TGFBR2 and used ITC to characterize the binding (Supplementary Fig. 9 and Supplementary Table 6).

To analyze the TGM6-D3 Arg[95]:TGFBR2 Asp[55] doubly hydrogen-bonded ion-pair (Fig. 7a), we substituted TGFBR2 Asp[55] or TGM6-D3

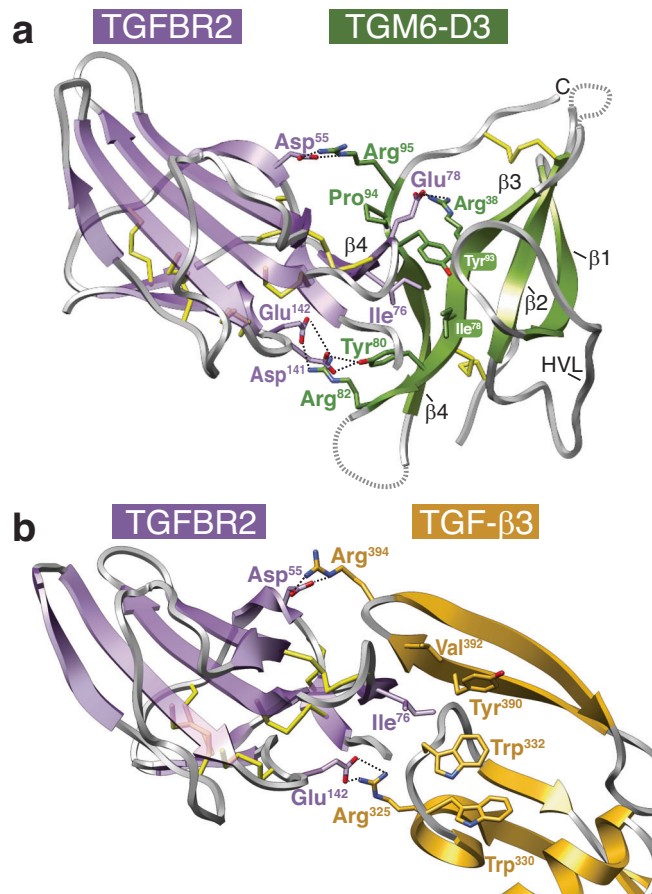

**Fig. 6 | Structure of the TGM6-D3:TGFBR2 complex and mimicry of mammalian TGF-β. a** Overall structure of the TGM6-D3:TGFBR2 complex determined by X-ray crystallography at a resolution of 1.4 Å. TGM6-D3 and TGFBR2 are shaded olive green and lavender, respectively. Loops that are not modeled due to weak density are indicated by dashed lines. Sidechains of key interfacial residues are shown, as are the intramolecular disulfide bonds. **b** Structure of the TGF-β3:TGFBR2 complex at a resolution of 2.15 Å (PDB 1KTZ). TGF-β3 and TGFBR2 are shaded burnt orange and lavender, respectively. Sidechains of key interfacial residues are shown, as are the intramolecular disulfide bonds. Structural data are available through the RCSB PDB under accession code 9E9G.

Arg[95] with alanine and found that these diminished the binding affinity by 6.4- or 11.6-fold (Supplementary Table 6). There is an additional nearby ion-pair between TGFBR2 Glu[78] positioned in the loop following β-strand 4 and TGM6-D3 Arg[38] on the tip of the HVL and substitution of TGM6-D3 Arg[38] with alanine diminished the binding affinity by 22-fold. If it is assumed the binding energy in this region of the TGM6-D3:TGFBR2 complex is a result of both the interactions described above, then the theoretical perturbation of simultaneously eliminating both interactions would be 140–255-fold. In the TGF-β1/-β3:TGFBR2 complex substitution of the residues that form the homologous doubly hydrogen-bonded ion-pair, TGFBR2 Asp[55] and TGF-β1/-β3 Arg[394], led to a 30–80-fold decrease in binding affinity (Fig. 7b and Supplementary Table 6). Thus, the energetic contribution of the two combined interactions in the TGM6-D3:TGFBR2 complex is comparable or exceeds the contribution of the single interaction in the TGF-β1/-β3:TGFBR2 complex.

The central hydrophobic interaction between TGFBR2 and TGM6-D3 was probed by substituting both TGFBR2 Ile[76] and TGM6-D3 Ile[78], Tyr[80], and Tyr[93] with alanine (Fig. 7c). These substitutions led to large perturbations, ranging from 21-fold for the TGFBR2 I76A substitution, to 15.4-, 134-, and 108-fold for the TGM6-D3 I78A, Y80A, and Y93A substitutions, respectively (Supplementary Table 6). In the

TGF-β3:TGFBR2 complex, substitution of Ile[76] with alanine led to a 12.5-fold reduction in affinity, while the effects of substituting of Trp[330], Tyr[390], and Val[392] between the fingers of the TGF-β1/-β3 were not reported due to the propensity of substitutions in this region of the protein to lead to misfolding[42] (Fig. 7d and Supplementary Table 6). The methyl and amide regions of the 1D ¹H NMR spectra of the TGM6-D3 I78A, Y80A, and Y93A variants have dispersed patterns similar to wild type, indicating that the effects of the substitutions on binding are not due to an overall perturbation of the folding (Supplementary Fig. 10). The central hydrophobic interaction clearly has an essential role for binding in the TGM6-D3:TGFBR2 complex, similar to that in the TGF-β1/-β3:TGFBR2 complex, however, it is not possible to directly compare the energetic contribution of these to binding due to the lack of TGF-β mutants and the multiple residues involved.

The functional significance of the TGM6-D3 Arg[82]:TGFBR2 Glu[142] hydrogen-bonded ion-pair was uncertain as there was defined, but weaker electron density for the Arg[82] guanidinium group, but little density for the sidechain Cβ and Cγ atoms and the backbone (Fig. 7e). The modeling of the Arg[82] guanidinium group is supported by omit maps in which Arg[82] and the subsequent three residues in the β3-β4 loop are absent (Supplementary Fig. 11); further, substitution of TGM6-D3 Arg[82], or its partner residue, TGFBR2 Glu[142], with alanine is shown to diminish the binding affinity by 11–12-fold (Supplementary Table 6). The adjacent hydrogen-bonding interaction between the phenolic hydroxyl of TGM6-D3 Tyr[80] and the sidechain carboxylate of Asp[141] also appears to be functionally significant as substitution of Asp[141] to alanine or TGM6-D3 Tyr[80] with phenylalanine reduced the affinity by 2.7- and 3.5-fold. Thus, similar to TGM6-D3 Arg[95]:TGFBR2 Asp[55] and TGM6-D3 Arg[38]:TGFBR2 Glu[78] ion-pairs on the opposite side of the interface, both interactions appear contribute and the theoretical perturbation of simultaneously eliminating both interactions is expected to be about 35-fold. In the TGF-β1/-β3:TGFBR2 complex substitution of the residues that form the homologous doubly hydrogen-bonded ion-pair, TGFBR2 Glu[142] and TGF-β1/-β3 Arg[325], led to a 12–30-fold decrease in the binding affinity (Fig. 7f and Supplementary Table 6). Thus, the combination of the hydrogen-bond and ion-pair interactions in TGM6-D3:TGFBR2 complex and hydrogen-bonded ion-pair in the TGF-β1/-β3:TGFBR2 complex contribute comparably to the overall binding energy.

## TGM6 function is dependent on high-affinity TGFBR2 binding, while TGM1 function is not

TGM1 is dependent upon both TGFBR1 and TGFBR2 for signaling, but it binds TGFBR2 moderately ($K_D$ 1.2–1.5 μM vs. 0.35 μM for TGM6) and single amino acid substitutions, such as Y253A that weaken binding by 40-fold or more, only modestly attenuate signaling[30]. In contrast, the Y253A substitution nearly abrogates signaling, even with very high concentrations of ligand, when the co-receptor binding domains, D45, are absent[27]. Thus, when there is trivalent engagement of receptors, TGM1 signaling activity is retained even if TGFBR2 binding affinity is significantly compromised, but when there is bivalent engagement of receptors, the same perturbation of TGFBR2 binding almost eliminates signaling activity.

This suggested that the identification of TGM1-D3 residues responsible for its lower affinity for TGFBR2 and swapping these into TGM6 would severely impair its inhibitory potential, owing to its bivalent engagement of receptors, TGFBR2 through D3 and a co-receptor through D45. The substitution of the corresponding TGM6-D3 residues into TGM1, by contrast, was expected to only modestly enhance signaling owing to its trivalent engagement of receptors.

To investigate this, we compared the amino acid sequences of TGM1-D3 and TGM6-D3, in which homologous residues differ in notation by the 160-amino acid length of D1–D2 present only in TGM1. We noted two differences that might be responsible for the reduced affinity

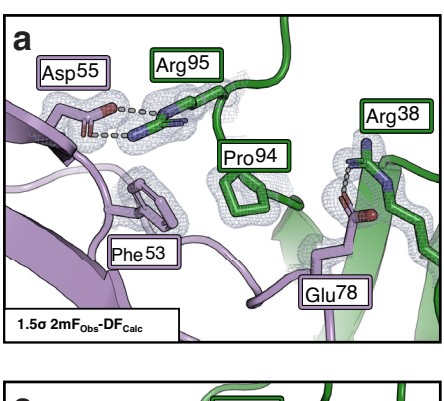

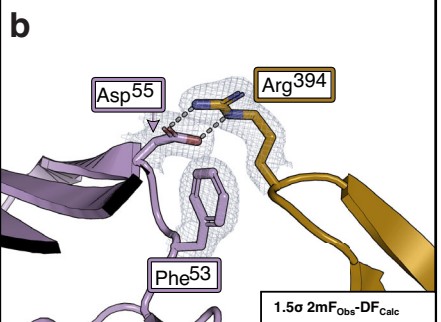

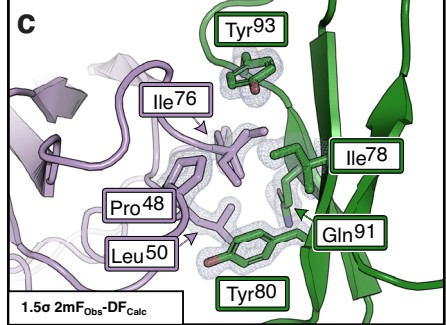

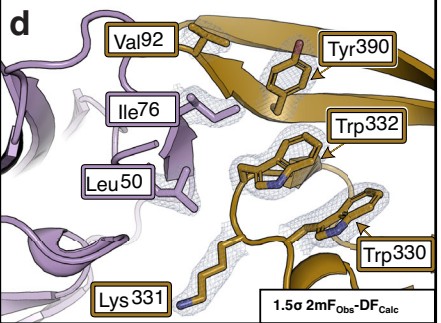

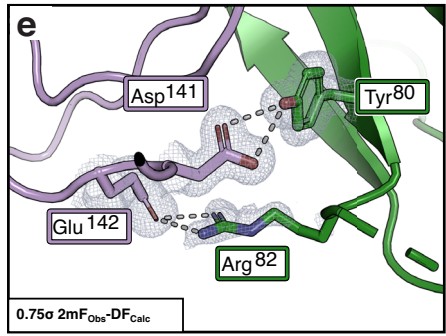

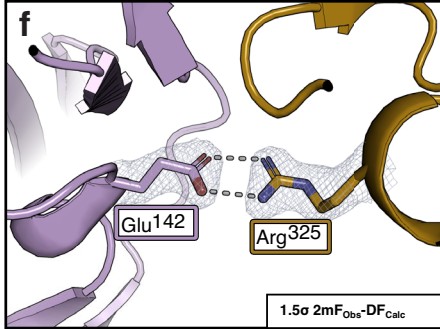

**Fig. 7 | Interface contacts of TGM6-D3 with TGFBR2 and mimicry of mammalian TGF-β.** Ionic interaction with TGFBR2 Asp[55] with Arg[95] of TGM6-D3 (**a**) and Arg[394] of TGF-β3 (**b**). TGFBR2 Glu[78] also has an ionic interaction with TGM6-D3 Arg[38], but it does not interact with TGF-β. Interaction of TGFBR2 Leu[50] and Ile[76] with the hydrophobic pocket on TGM6-D3 formed by Tyr[93], Ile[78], and Tyr[80] (**c**) and TGF-β3 formed by Tyr[390], Val[392], and Trp[330] (**d**). Ionic interaction with TGFBR2 Glu[142] and Asp[141] with Tyr[80] and Arg[82] of TGM6-D3, respectively (**e**) and TGFBR2 Glu[142] with Arg[325] of TGF-β3 (**f**). In all panels, TGFBR2 is shaded lavender and sidechains of key interaction residues are shown. In panels (**a**), (**c**), and (**e**), TGM6 is shaded green, and, in panels (**b**), (**d**), and (**f**), TGF-β3 is shaded burnt orange. Structural data are available through the RCSB PDB under accession code 9E9G.

of TGM1 for TGFBR2 (Fig. 1d): (1) substitution of the Lys[254]–Asn[255] (KN) dipeptide in TGM1 in place of the Pro[94]–Arg[95] (PR) dipeptide in TGM6, and (2) substitution of the Lys[241]–Ser[242]–Gly[243]–Thr[244] (KSGT) tetrapeptide in TGM1 in place of the Gln[81]–Arg[82]–Arg[83]–Gly[84] (QRRG) tetrapeptide in TGM6. The Lys[254] or Lys[241] of TGM1 KN or KSGT peptides may not functionally replicate the hydrogen-bonded ion-pairs of TGM6 Arg[95] or Arg[82] with TGFBR2 Asp[55] or Glu[142], respectively, leading to weaker binding and impairment of function (Fig. 7a, e).

The first of these was tested by substituting the PR dipeptide of TGM6-D3 with the KN dipeptide from TGM1, and vice versa, and measuring the binding affinity of the chimeric proteins, TGM6-D3 KN and TGM1-D3 PR, for TGFBR2 using ITC (Supplementary Fig. 9). Unexpectedly, the affinities of TGM1-D3 PR and TGM6-D3 KN for TGFBR2 were either slightly weaker or indistinguishable from the parental wild-type proteins, indicating that the lysine of the KN dipeptide of TGM1 can functionally replicate the TGM6-D3 Arg[95]:TGFBR2 Asp[55] interaction (Supplementary Table 6). The second was tested by exchanging the KSGT tetrapeptide in TGM1 with the QRRG tetrapeptide in TGM6 and vice versa and measuring the affinity

of these chimeric proteins for TGFBR2 using ITC (Fig. 8a, b). These showed that binding of TGM6-D3 KSGT to TGFBR2 was impaired by 12.5-fold compared to wild-type TGM6-D3, while binding of TGM1-D3 QRRG to TGFBR2 was enhanced by ninefold compared to wild-type TGM1-D3, indicating that these residue differences indeed underlie the differential affinity of TGM1-D3 and TGM6-D3 for TGFBR2. To ascertain whether the replacement of TGM6-D3 Arg[82] with the corresponding residue of TGM1-D3, Ser[242], and vice versa was responsible for the loss and gain of affinity, we generated the TGM6-D3 R82S and TGM1-D3 S242R single amino acid variants and measured their affinity for TGFBR2 using ITC (Supplementary Fig. 9 and Supplementary Table 6). The loss and gain of binding affinity compared to wild type were more moderate compared to the tetrapeptide swaps, 8.6-fold loss for TGM6-D3 R82S and 4.0-fold gain for TGM1-D3 S242R, but they were also more aligned with the overall affinity difference between TGM1 and TGM6 for TGFBR2, indicating that these alone are responsible for the affinity difference.

To determine how the loss or gain of TGFBR2 binding affinity affected function, we began by comparing the inhibitory activity of

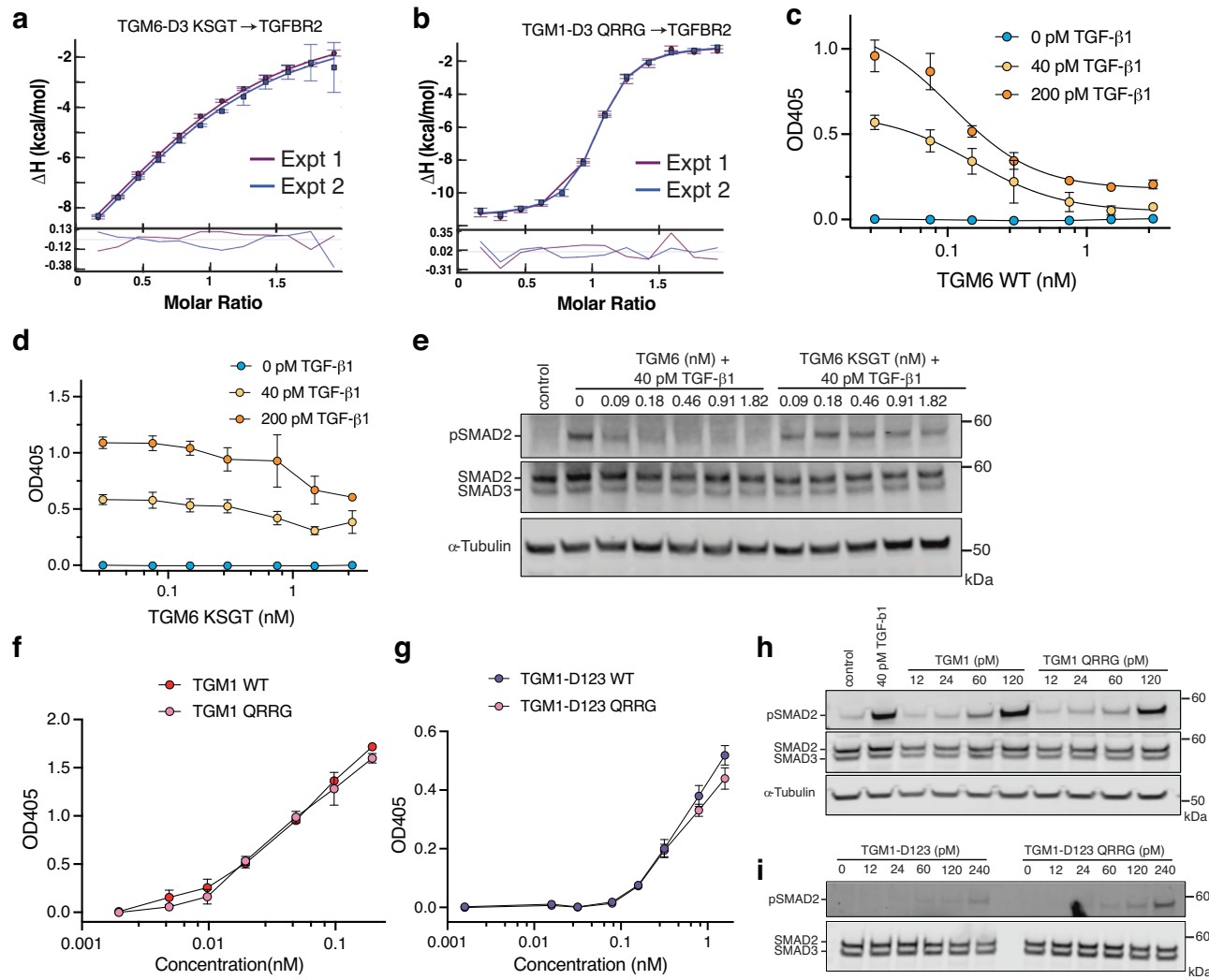

**Fig. 8 | TGM6 residues responsible for its high affinity for TGFBR2.** ITC fitted binding isotherms (top) and residuals (bottom) for titration of TGM6-D3 KSGT (**a**) or TGM1-D3 QRRG (**b**) into TGFBR2. Fitted binding isotherms are overlaid as two experiments shaded purple and blue. Inhibition of TGF-β1 signaling in MFB-F11 fibroblasts as detected by the conversion of *p*-nitrophenylphosphate by secreted alkaline phosphatase (**c**, **d**) or by pSMAD2 western blotting (**e**) by either TGM6 (**c**, **e**) or TGM6 KSGT (**d**, **e**). In the MFB-F11 assay, the cells were stimulated with 0, 40, or 200 pM TGF-β1 (blue, yellow, and orange symbols, respectively). Activation of TGF-β1 signaling in MFB-F11 fibroblasts as detected by the conversion of *p*-nitrophenylphosphate by secreted alkaline phosphatase (**f**, **g**) or by pSMAD2 western blotting (**h**, **i**) by either TGM1 or TGM1 QRRG (**f**, **h**) or by TGM1-D123 or TGM1-D123 QRRG (**g**, **i**). Data shown in (**c**, **d**, **f**, **g**) are mean and standard deviation of triplicate measurements from one experiment. Blots shown in (**e**, **h**, **i**) are from one experiment with α-tubulin serving as a loading control. Source data of (**a**, **b**) and (**c**, **i**) provided through Figshare [https://doi.org/10.6084/m9.figshare.28179359] or the Source Data file associated with the article, respectively.

TGM6 or TGM6 KSGT in MFB-F11 fibroblasts as assessed by either its secreted alkaline phosphatase reporter activity or induction of pSMAD2/3. In both assays TGM6 KSGT failed to inhibit induction of signaling by either TGF-β1 or TGM1, except at the highest concentration where it only partially inhibited, while TGM6 WT inhibited at all concentrations tested (Fig. 8c–e and Supplementary Fig. 12). We then compared the induction of signaling in MFB-F11 fibroblasts by TGM1 or TGM1-D123, either as the wild-type protein or the QRRG variant. In the reporter assays, we observed no apparent difference between the wild-type protein or QRRG variant, regardless of whether the full-length or truncated protein was assayed (Fig. 8f, g), while in the pSMAD2/3 assays, we observed no apparent differences between the wild-type protein or QRRG variant in the context of full-length TGM1 (Fig. 8h), but a small but consistent increase in potency at the three highest concentrations tested in the context of the truncated protein (Fig. 8i). Thus, consistent with expectations, swapping the residues responsible for the lower affinity of TGM1 for TGFBR2 into TGM6 dramatically impaired its inhibitory activity, whereas swapping the residues

responsible for the greater affinity of TGM6 for TGFBR2 into TGM1 led to only modest enhancements in signaling activity.

## Discussion

The studies of TGM6 reported here further contribute to our understanding of the adaptations of the TGM family of proteins. The structure of TGM6-D3 bound to TGFBR2 reveals the remarkable molecular mimicry that enables D3 to bind TGFBR2 in a manner that closely resembles TGF-β1/-β3[38,39]. In the structure of the TGM6-D3:TGFBR2 complex, we find that not only does the parasite protein engage TGFBR2 in the same overall manner as TGF-β1/-β3 through a central hydrophobic cleft and flanking hydrogen-bonded ion-pairs, but the presentation of key interacting residues, such as Arg[95] and Arg[82], which form hydrogen-bonded ion-pairs with TGFBR2 Asp[55] and Glu[142], are also remarkably similar to those of TGF-β1/-β3. Though the hydrogen-bonded ion-pairs that TGM6-D3 Arg[95] and Arg[82] have with Asp[55] and Glu[142] of TGFBR2 do not contribute as much energetically as the corresponding interactions in the TGF-β1/-β3 complex, this is

compensated by additional hydrogen-bonds and hydrogen-bonded ion-pairs, including TGM6-D3 Arg[38]:TGFBR2 Glu[78] and TGM6-D3 Tyr[80]:TGFBR2 Asp[141], that are lacking in the TGF-β1/-β3:TGFBR2 complex. The additional interactions in the TGM6-D3:TGFBR2 complex arise in part from adaptations of the TGM-D3 CCP fold, such as the structurally ordered HVL[27], that provide a greater number of opportunities to position residues that can productively interact with residues on TGFBR2.

This study has also shown that the three-domain TGM6 functions as a potent inhibitor of TGF-β and TGM1 signaling. The paradox of why the adult parasite might have evolved to co-express inhibitory TGM6 alongside signaling-activating TGMs likely stems from the different cell populations they target, with TGM6 potently inhibiting TGF-β and TGM1 signaling in fibroblasts and epithelial cells, but not splenic T cells, the cell type targeted by TGM1[30]. The targeting of TGM6 to fibroblasts and epithelial cells is likely mediated by binding of D45 to a co-receptor that is present on fibroblasts and epithelial cells, but not T cells. This is suggested by the finding that D3 alone is not inhibitory on T cells, the inhibitory potency of the full-length protein on fibroblasts and epithelial cells is 1300-fold lower than the binding affinity of D3 for TGFBR2, and D45 does not bind CD44.

The targeting of TGM6 to fibroblasts might reflect a strategy the parasite has adopted to reduce fibrotic activity as it transitions through its life cycle, which involves invasion of newly arriving larvae through the intestinal epithelium to encyst in the muscle. Upon maturing in the intestinal submucosa, the adult parasites burrow back through the intestinal wall to the lumen where they mate and produce eggs that are released to the environment in the feces. In light of the considerable tissue damage that would occur in this process, and the well-established role of TGF-β and TGM1 in contributing to wound repair by stimulating deposition of type I collagen[6,16,43] and driving tissue fibrosis if signaling is dysregulated[16], it is possible that TGM6 modulates TGF-β signaling in fibroblasts, which would reduce the degree of collagen deposition and fibrosis. In addition, TGM6 may have roles in inhibiting TGF-β signaling in other cell types to minimize adverse activity of TGM agonists on cell types aside from those responsible for host immunomodulation.

The TGF-βs, in addition to stimulating pro-tolerogenic signaling that is required for immune homeostasis, are highly pleiotropic and induce signaling that is important for homeostatic control of many other essential processes[44]. The targeting of the TGF-β pathway by the helminth H. polygyrus therefore has potentially adverse consequences for the host. To overcome potentially deleterious off-target effects, and thus limit damage to the host, the parasite has adapted the multidomain CCP scaffold proteins. These can bind TGFBR1 and TGFBR2 to activate the TGF-β pathway, but are targeted to relevant immune subsets, such as T lymphocytes and myeloid cells, by binding co-receptors such as CD44 and CD49d[6,30,31]. They can also bind TGFBR2 and a co-receptor, but not any other signaling receptor of the TGF-β family, to simultaneously antagonize signaling in cell populations, such as fibroblasts and epithelial cells, that could be detrimental to host fitness if the TGF-β pathway is activated.

The affinities of the relevant receptor binding domains of the TGM proteins have also evolved to confer high target specificity. In agonists that bind TGFBR1, TGFBR2, and a co-receptor, such as TGM1 and TGM4, this has been achieved by weakening binding to TGFBR2 and by enhancing binding to the relevant co-receptors present on different immune cell subsets[27,30,31]. In TGM1 and TGM4, the full-length proteins signal with efficacies that are comparable to the native cytokine for their target cell populations, in spite of TGFBR2 binding affinities that are either moderate (ca. 1 μM, TGM1) or weak (ca. 50–100 μM, TGM4), while truncated forms, lacking the co-receptor binding domains, either signal moderately or not at all. In antagonists, such as TGM6 that bind TGFBR2 and a co-receptor, it is evidently not possible to weaken TGFBR2 binding to the same extent as the agonists, though it is notable that the TGFBR2 binding affinity of TGM6 is about 10-folder weaker than that of TGF-β1/-β3 for TGFBR2 (ca. 300 nM for TGM6 vs. ca. 30 nM for TGF-β1/-β3[23,39]), that its inhibitory potential is lost if binding to TGFBR2 is only moderately impaired, and it is only able to antagonize when also bound to a co-receptor through D45. This apparent moderation of the TGFBR2 binding affinity may allow TGM6 to discriminate between target populations according to their co-receptor profile, enabling effective antagonism in fibroblasts and epithelial cells, but not immune cells, the targets of TGM agonists.

The TGF-β pathway has arisen as an important target for the development of antagonists for cancer immunotherapy[45] and for fibrotic disorders[16], but also for agonists for treating autoimmune disorders, such as inflammatory bowel disease[17]. Though many approaches have been proposed for targeting the TGF-β pathway with antagonists and agonists, no agents have been approved for use in humans[46,47]. The co-receptor-dependent targeting of the TGF-β pathway with both agonists and antagonists by the helminth H. polygyrus provides both a vivid illustration of pathogen evolutionary innovation, and an instructive template for the development of effective therapies for targeting the cancer and fibrosis-promoting activities of the TGF-βs in humans.

## Methods

### Expression and purification of TGM proteins

The amino acid sequences of the TGM proteins used in this study are presented in Supplementary Table 7. The plasmids used to produce TGM1, TGM1-D123, and TGM6 in mammalian cells were previously described[6,28]. The plasmids used to produce TGM1-D3, TGM6-D3, and TGM6-D45 in bacteria were generated by inserting the corresponding coding sequences downstream of the thrombin cleavage site in a modified form of pET32a (EMD-Millipore, Cat# 69015-3) with a His$_{10}$ tag instead of His$_6$.

TGM1, TGM1-D123, and TGM6 were expressed in expi293 cells (Invitrogen, Cat# A14527) and purified by binding the protein in the conditioned medium onto a HisPur (Thermo, Cat# A50590) immobilized metal affinity column (IMAC). The bound protein was eluted using a 0–0.5 M imidazole gradient. The eluted protein peak was concentrated, deglycosylated with a 14 h incubation with PNGAse F at 30 °C, and further purified by size exclusion chromatography (Cytiva HiLoad 26/60 Superdex 75, Cat# 28989333).

TGM1-D3 was overexpressed in E. coli at 37 °C in the form of insoluble inclusion bodies, refolded, and purified as described[27]. TGM6-D3 and TGM6-D45 were produced and purified similarly, with two exceptions: (1) the lysis supernatant for TGM6-D3 was retained and combined with the urea-solubilized inclusion bodies prior to purification by nickel metal affinity chromatography and (2) rather than final purification on a Source Q column, both were purified on a Source 15S column (Cytiva, Cat# 17094401). TGM6-D3 was bound in 25 mM sodium acetate, 2 M urea, 10 μM leupeptin hemisulfate, 10 μM pepstatin, 100 mg L$^{-1}$ benzamidine, pH 4.8 and eluted with a 0–0.35 M NaCl gradient, while TGM6-D45 was bound in 25 mM Tris, 2 M urea, 10 μM leupeptin hemisulfate, 10 μM pepstatin, 100 mg L$^{-1}$ benzamidine pH 7.5, and eluted with a 0–0.2 M NaCl gradient.

### Expression and purification of type I receptors

The amino acid sequences of the extracellular domains of the type I receptors used in this study are presented in Supplementary Table 8. The plasmids used to produce ALK1, ALK3, and ALK5 (also known as TGFBR1) in bacteria were previously described[48,49]. The plasmids used to produce ALK2 and ALK4 in bacteria were constructed by inserting the corresponding coding sequence downstream of the thrombin cleavage site in either pET32a or pET15b (EMD-Millipore, Cat # 69661-3).

ALK1, ALK3, and ALK5 were expressed in E. coli grown on M9 minimal medium with ¹⁵N-labeling, refolded from urea-solubilized

inclusion bodies, and purified as described[48,49]. ALK4 was expressed, refolded, and purified similarly to ALK1, with the exception that the refolded protein was purified in two steps, first by loading onto a Source 15Q column (Cytiva, Cat# 17094701) in 25 mM Tris, 2 M urea, 10 μM leupeptin hemisulfate, 10 μM pepstatin, and 100 mg L$^{-1}$ benzamidine pH 8.0 and eluting with a 0–0.35 M NaCl gradient, and second by loading onto a C18 semi-preparative reverse phase column (Phenomenex Jupiter 5 μm C18 300 Å, Cat# 00G-4053-N0) and eluting with a 5%–70% acetonitrile gradient.

ALK2 was expressed on minimal medium with $^{15}$N labeling at 37 °C until the $A_{600}$ reached 0.4, followed by transfer to 14 °C and induction of expression with IPTG when the $A_{600}$ reached 0.6. $^{15}$N-ALK2 protein was purified from the lysis supernatant by loading onto IMAC column and eluting with a 0–0.5 M imidazole gradient. Fractions containing $^{15}$N-ALK2 were pooled, thrombin treated, dialyzed against 25 mM CHES, pH 9.0, and purified in two steps, first by loading onto a Source Q column and eluting with a 0–0.35 M NaCl gradient, and second by loading onto a C18 semi-preparative reverse phase column and eluting with a 5%–70% acetonitrile gradient.

### Expression and purification of type II receptors

The amino acid sequences of the extracellular domains of the type II receptors used in this study are presented in Supplementary Table 9. The plasmid used to produce TGFBR2 in bacteria was previously described[50]. The plasmids used to produce ActRII and ActRIIB in bacteria were constructed by inserting the corresponding coding sequence downstream of the N-terminal His$_6$ tag and thrombin cleavage site in pET15b. The plasmid used to produce BMPRII extracellular domain in mammalian cells was constructed by inserting the coding sequence for this downstream of the rat serum albumin signal peptide, a His$_6$ tag, and a thrombin cleavage site in a modified form of pcDNA3.1+ (Invitrogen, Cat# V79020).

TGFBR2 was overexpressed in *E. coli* at 37 °C in the form of insoluble inclusion bodies, refolded, and purified as described previously[50]. ActRII and ActRIIb were expressed in the form of insoluble inclusion bodies in *E. coli* BL21(DE3) cells (EMD-Millipore, Cat# 69450) and were refolded and purified similarly to ALK1, with the exception that the refolded protein was purified by loading onto a Source Q column in 25 mM sodium phosphate, 10 μM leupeptin hemisulfate, 10 μM pepstatin, and 100 mg L$^{-1}$ benzamidine pH 6.6 and eluting with a 0–0.35 M NaCl gradient. BMPRII was expressed in expi293 cells (Invitrogen) and purified from the conditioned medium using nickel metal affinity chromatography and SEC in the manner described above for TGM6.

### Expression and purification of mmTGF-β2-7M2R and mCD44

The amino acid sequences of mmTGF-β2-7M2R and mCD44 used in this study are presented in Supplementary Table 10. mm-TGF-β2-7M2R was overexpressed in *E. coli* BL21(DE3) cells at 37 °C in the form of inclusion bodies, refolded, and purified[34]. mCD44 was produced in expi293 suspension cultured mammalian cells and purified[30].

### Point mutants and validation of recombinant proteins

Single amino acid mutants of TGFBR2, TGM6-D3, or TGM1-D3 were generated by site-directed mutagenesis[51]. Multiple amino mutants of TGM1 and TGM6 were generated by gene synthesis (Twist Biosciences). Coding sequences of all wild-type and mutant proteins were verified by DNA sequencing over the length of their coding sequences. Masses of all recombinant proteins produced in *E. coli*, including all point mutants, were confirmed by liquid chromatography–electrospray ionization time-of-flight mass spectroscopy (Micro TOF, Bruker).

### NMR data collection

Samples of $^{15}$N TGM6-D3 and its complex with TGFBR2 were prepared at a concentration of 150 μM in 25 mM Na$_2$HPO$_4$, 10 μM leupeptin hemisulfate, 10 μM pepstatin, 100 mg L$^{-1}$ benzamidine, 0.05% (w/v) NaN$_3$, 5% $^2$H$_2$O, pH 5.5. Samples of $^{15}$N ALK1, $^{15}$N ALK2, $^{15}$N ALK3, $^{15}$N ALK4, and $^{15}$N ALK5 and their corresponding samples containing 1.125 molar equivalents of TGM6-D45 or the TGM6:TGFBR2 binary complex were prepared at a concentration of 100 μM $^{15}$N-labeled receptor in 25 mM Na$_2$HPO$_4$, 10 μM leupeptin hemisulfate, 10 μM pepstatin, 100 mg L$^{-1}$ benzamidine, 0.05% (w/v) NaN$_3$, 5% $^2$H$_2$O, pH 6.0.

All NMR samples were transferred to 5 mm susceptibility-matched microtubes for data collection (Sigma-Aldrich). NMR data were collected at 303.15 K using 600, 700, or 800 MHz Bruker NMR spectrometers equipped with 5 mm $^1$H{$^{13}$C,$^{15}$N} z-gradient "TCI" cryogenically cooled probes running TopSpin 3.5, 2.1, or 3.1 (Bruker Biospin, Billerica, MA). 2D $^1$H–$^{15}$N HSQC spectra were recorded with sensitivity enhancement[52], water flip-back pulses[53], and WATERGATE water suppression pulses[54]. NMR data were processed using NMRPipe 9.9[55] and analyzed using NMRFAM-SPARKY 1.2_3.115[56].

### ITC measurements

ITC data were generated using a Microcal PEAQ-ITC instrument running version 1.40 of the Malvern PEAQ-ITC control software (Malvern Instruments, Westborough, MA). All experiments were performed in ITC buffer (25 mM HEPES, 150 mM NaCl, 0.05% NaN$_3$, pH 7.4). The proteins in the syringe and sample cell and their concentrations are provided in the respective data tables. Prior to each experiment, all proteins were dialyzed three times against ITC buffer and were concentrated or diluted as necessary before being loaded into the sample cell or syringe. For each experiment, either thirteen 3.0 μL or nineteen 2.0 μL injections were performed with an injection duration of 4 s, a spacing of 150 s, and a reference power of 10. Integration and data fitting were performed using Nitpic 2.1.0[57] and Sedphat 15.2b[58,59]. No more than two outlier data points were removed from any one ITC data set for analysis. The TGM6:TGFBR2 binding experiment was globally fit to a simple binding model from two experimental replicates. The TGM6-D3:TGFBR2 binding experiment was globally fit to a simple binding model from three experimental replicates. The TGM6-D3 variant and TGFBR2 variant binding experiments were fit to a simple binding model from 1–2 replicates per variant. Competition experiments were performed with TGFBR2 in the syringe and the competitors in the sample cell (Supplementary Table 3). The data were globally fit using a simple competitive binding model with one replicate per condition.

### X-ray structure determination

TGM6-D3 (residues 16–102 of the full-length construct) and TGFBR2 46–155 were mixed in a 1.1-to-1.0 ratio, with TGM6-D3 being in slight excess. The binary complex was fractionated by SEC using a HiLoad Superdex 75 26/60 column (Cytiva, Cat# 28989334) in 25 mM Tris, 100 mM NaCl, 0.05% NaN$_3$, pH 8.0. The fractions containing the binary complex were pooled and concentrated to 50 mg mL$^{-1}$ for crystallization. The binary complex was crystallized in 0.1 M sodium cacodylate, 25% (w/v) PEG 4000, pH 6.5. Large star-burst-like crystal clusters with plate-like arms grew at ambient temperature in about 3 days.

Harvested crystals were briefly soaked in mother liquor containing 14% glycerol for cryoprotection and mounted in nylon loops with excess mother liquor wicked off. The looped crystals were then flash-cooled in liquid nitrogen prior to data collection. Data were collected at the Southeast Regional Collaborative Access Team (SER-CAT) 22-ID beamline at the Advanced Photon Source, Argonne National Laboratory using SERGUI beamline control and data collection software. Data were integrated and scaled using XDS Ver. Jun 30, 2023[60]. The structure was determined by the molecular replacement method implemented in PHASER 2.7[61] using the 1.1 Å TGFBR2 X-ray structure (PDB 1M9Z)[37] and the TGM1-D3 NMR structural ensemble (PDB 7SXB)[27] as search models. Coordinates were refined using Phenix.refine 1.20.1[62]

and alternated with manual rebuilding using COOT 0.9.8.7[63]. Molecular graphics and analyses performed with Pymol Open Source Ver. 2.6 (Schrödinger, LLC) and UCSF ChimeraX Ver. 1.5[64]. Data collection and refinement statistics are shown in Supplementary Table 5. Final structure factors and model are deposited in the RCSB PBD under accession number 9E9G.

## TGF-β/TGM inhibition assays in NIH-3T3 fibroblasts

TGF-β/TGM inhibition assays utilizing NIH-3T3 cells were performed using NIH-3T3 cells stably transfected with a CAGA$_{12}$-luciferase reporter construct as previously reported[33]. Briefly, NIH-3T3-CAGA cells were plated at a concentration of $2 \times 10^4$ cells per well in 24-well plates containing Dulbecco's modified Eagle's medium (DMEM) plus 10% fetal calf serum (FCS) and allowed to attach for 18 h. Cells were washed with PBS and incubated with DMEM plus 0.1% FCS for 6 h. After this initial incubation, increasing concentrations of either TGM6 or TGM6-D3 were added to the wells for 30 min prior to stimulation with 0.1 ng/mL TGM1 or TGF-β1 (Thermo Peprotech, Cat# 100-21C). The cells were incubated for 15 h and then washed with PBS and lysed using 100 μL of reporter lysis buffer (Promega, Cat#E4030). To measure luciferase activity, 30 μL Luciferase Assay Reagent (Promega, Cat# E1483) was added to 20 μL of lysate. The protein concentration of each lysate was analyzed using Bio-Rad protein assay reagent (Cat# 5000006) according to the manufacturer's instructions. Luciferase units obtained were normalized to the protein content of each well. All experiments were performed with three independent wells per condition and the experiments were repeated at least twice. Cells were routinely screened for mycoplasma.

## TGF-β/TGM inhibition and activation assays in MFB-F11 fibroblasts

TGF-β/TGM inhibition assays utilizing MFB-F11 cells containing a TGF-β-responsive alkaline phosphatase reporter[32] were performed as previously reported[6]. Briefly, 80%–90% confluent cells were detached with trypsin, and resuspended in DMEM containing 2% FCS, 100 U mL$^{-1}$ penicillin, 100 mg mL$^{-1}$ streptomycin, and 2 mM L-glutamine at a concentration of $8 \times 10^5$ cells mL$^{-1}$. Cells were plated at $4 \times 10^4$ (50 μL) cells per well of a 96-well flat-bottomed plate and left to incubate at 37 °C for 2 h. After this initial incubation, increasing concentrations of either full-length TGM6 or TGM6-D3 were added to the wells in a volume of 25 μL. After 30 min, cells were stimulated with 40 or 200 pM TGF-β or TGM1 in a volume of up to 25 μL and incubated for another 24 h at 37 °C, 5% CO$_2$. The final volume in each well was 100 μL. After the second incubation, 20 μL of supernatant was aspirated from each well, added to a 96-well flat-bottomed plate. Sigma Fast™ p-nitrophenylphosphate substrate (EMD-Millipore, Cat# N2770) was reconstituted in sterile Milli-Q water and 180 μL of it was added to each well of the 96-well plate and incubated at room temperature in the dark for up to 24 h. Plates were read at 405 nm on an Emax precision microplate reader (Molecular Devices, San Jose, CA). All conditions were set up in triplicate and repeated at least twice. IC$_{50}$ values were calculated in Prism 9 (GraphPad Software, Boston, MA) by globally fitting the replicates of each inhibition assay to a nonlinear dose-response inhibition model.

For TGM1 activation assays, MFB-F-11 cells were seeded at $4 \times 10^4$ (50 μL) cells per well of a 96-well flat-bottomed plate and left to incubate at 37 °C for 4 h. Increasing concentrations of full-length TGM1 or TGM1-D123 (wild-type or QRRG mutant) were added to the wells to final volume of 100 μL and incubated for another 24 h at 37 °C, 5% CO$_2$. The remainder of the assay was completed as described above. Cells were routinely screened for mycoplasma.

## pSMAD stimulation and western blotting

MFB-F11 cells were cultured in 6-well tissue culture plates until they reached a confluency of 80%–90% in complete growth medium (DMEM, 10% FBS, 1% L-glutamine, 1× penicillin–streptomycin). The growth medium was then replaced with serum-free DMEM, and the cells were incubated at 37 °C with 5% CO$_2$ for 4 h. To stimulate pSMAD2, TGFβ, TGM1, or TGM1-D123 were added to the cells and incubated at 37 °C for 1 h. To inhibit pSMAD2, the cells were incubated with increasing concentrations of TGM6 for 30 min. Following this incubation, the cells were stimulated with either TGFβ or TGM1 for 1 h at 37 °C. The cells were washed with ice-cold PBS and lysed with RIPA buffer (0.05 M Tris-HCl, pH 7.4, 0.15 M NaCl, 0.25% deoxycholic acid, 1% NP-40, 1 mM EDTA) containing 1X Halt protease and phosphatase inhibitor cocktail (Invitrogen, Cat# 78440). Cell lysates were cleared by centrifugation at $13,000 \times g$, 4 °C for 5 min, and protein concentrations were estimated using the Precision Red reagent (Cytoskeleton Inc, Cat# ADV02). Protein samples were prepared by mixing 1X LDS (Invitrogen, Cat#NP0007), 25 mM DTT, and boiling at 100 °C for 5 min. Equal concentrations of cell lysates were analyzed on 4%–12% Bis-Tris NuPAGE SDS-PAGE gels (Invitrogen, Cat#NP0322) and transferred onto a nitrocellulose membrane using the iBlot2 system (Invitrogen, Waltham, MA). The membranes were treated with a 5% non-fat milk blocking solution for 1 h and incubated with the primary antibody (diluted 1:1000 in 5% BSA-containing TBST) overnight at 4 °C. The antibodies used for blotting were: phospho-Smad2 antibody (Cell Signaling Technologies, Cat# 8828), total Smad2/3 antibody (Cell Signaling Technologies, Cat# 3102), and alpha-tubulin antibody (Abcam, Cat# ab7291). The blots were then washed three times (5 min each) with 1X TBST. To detect the protein bands, a fluorescently conjugated secondary antibody (diluted 1:10,000 in 5% BSA-containing TBST) was used, and the bands were visualized using the Odyssey CLx Imaging System with LICOR imaging studio 5.0 (LI-COR Biosciences, Lincoln, NE). Uncropped blots are provided as source data.

## Foxp3$^+$ Treg induction assay

Induction of the Foxp3 transcription factor in murine splenic CD4+ T cells was performed[65]. Briefly, a single cell suspension was prepared from the spleens of naïve Foxp3-GFP BALB/c transgenic mice[66], with 2 min incubation in red blood cell lysis buffer (Sigma), then washed and resuspended in RPMI containing HEPES, supplemented with 2 mM L-glutamine, 100 U/mL of penicillin and 100 μg/mL of streptomycin, 10% heat-inactivated FCS (Gibco, Cat# A5209502), and 50 nM 2-mercaptoethanol. Naive CD4+ T cells were isolated by magnetic sorting using the mouse naïve CD4+ T-cell isolation kit (Miltenyi, Cat# 130-104-453) on the AutoMACS system (Miltenyi, Auburn, CA) as per the manufacturer's instructions. Cells were cultured at $2 \times 10^5$ per well in flat-bottomed 96-well plates with the addition of IL-2 (Miltenyi, Cat# 170-076-147) at a final concentration of 400 U/mL and pre-coated with 10 μg/mL of anti-CD3 (eBioscience, Cat# 14-0031-86). Cells were cultured with 5 ng/mL TGF-β1 or TGM1 with or without increasing concentrations of TGM6 at 37 °C in 5% CO$_2$ for 72 h before being removed for flow cytometric analysis. Briefly, cells were washed with PBS and stained with Fixable Viability Dye eFluor 506 (Invitrogen, Cat#65-0866-14) and TruStain FcX PLUS anti-mouse CD16/32 antibody (Biolegend, Cat# 156604) to identify live cells and prevent unspecific binding, respectively. Following this, cells were incubated with fluorochrome-conjugated anti-CD4 (Biolegend, Cat# BV650). Cells were analyzed using a BD FACSCelesta Cell Analyzer (BD, Lakes, NJ) and data were analyzed by FlowJo software (BD, Lakes, NJ). Analysis of stained cells was performed on single live cells.

## Transcriptional florescent protein-based reporter assays

The CAGA-MLP-dynGFP lentiviral vector was used for fluorescent protein-based reporter assays[67]. The BRE-MLP-mCHERRY-d2 lentiviral reporter was made using the pGL3-MLP-BRE-Luc plasmid (cloning information available on request)[68]. Lentiviruses were generated by transfection HEK293T cells with packaging constructs and the lentiviral constructs using standard protocols. Cells were exposed to

lentivirus containing pLV-CAGA-MLP-dynGFP or pLV-BRE-MLP-mCHERRYd2 for 48 h, after which the cells were selected using puromycin (CAGA-MLP-dynGFP) or blasticidin (BRE-MLP-mCHERRYd2). Murine NIH-3T3 fibroblasts containing either the CAGA-MLP-dynGFP or the BRE-MLP-mCHERRYd2 reporter, or murine NM-18 epithelial cells containing the CAGA-MLP-dynGFP reporter, were seeded in 96-well plates. For the CAGA-MLP-dynGFP reporter, the cells were stimulated the next day with either TGF-β3 or activin A in 10% serum. For the BRE-MLP-mCHERRYd2 the cells were put on media without serum overnight and were subsequently stimulated with BMPs. Directly after ligand stimulation, the cells were placed in the IncuCyte S3 live-cell imaging analysis system (Sartorius). The cells were imaged every 3 h for a period of 48 h. Fluorescence intensity was analyzed using the IncuCyte software. Cells were routinely screened for mycoplasma.

### Statistical analyses

Statistical analyses were performed using Prism 10 (GraphPad Software, Boston, MA), Nitpic 2.1.0[57], and Sedphat 15.2b[58,59], as appropriate. For comparisons of two groups, a Student's unpaired two-sided $t$-test was used, assuming unequal variance. $P$ values of <0.05 were considered statistically significant. Sample sizes were chosen empirically based on the laboratory's previous experience in the calculation of experimental variability; sample sizes for each experiment were not pre-determined by individual power calculations.

### Reporting summary

Further information on research design is available in the Nature Portfolio Reporting Summary linked to this article.

## Data availability

The X-ray source data used in this study are available in the RCSB PDB database under accession code 9E9G. The NMR and ITC source data have been deposited in the Figshare repository [https://doi.org/10.6084/m9.figshare.28179359]. Any additional information required to reanalyze the data reported in this paper is available from the lead contacts upon request. Plasmids generated in this study are maintained in the laboratories of Andrew Hinck (ahinck@pitt.edu), Richard M. Maizels (rick.maizels@glasgow.ac.uk), Peter ten Dijke (P.ten_Dijke@lumc.nl), and Gareth Inman (Gareth.Inman@glasgow.ac.uk) and will be made available upon request. Source data are provided with this paper.

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

## Acknowledgements

We would like to thank Chang Byeon for his assistance in producing and purifying proteins used in the binding studies and Matthew Whitley for his assistance with analysis of the diffraction data. The manuscript was critically reviewed by Catherine Winchester (CRUK Scotland Institute). This research was supported by the NIH through an RO1 (GM58670) and RO3 (AI53915) awarded to A.H. and an F30 (AI157069) awarded to A.M. and the Wellcome Trust through an Investigator Award (Ref 219530) to R.M.M., Discovery Award to P.tD., R.M.M. and A.P.H. (Ref 306173), and the Wellcome Trust core-funded Wellcome Centre for Investigative Parasitology (Ref 104111). C.S., J.P., and G.J.I. were supported by CRUK core funding to the CRUK Scotland Institute (A31287) and to G.J.I. (A29802). X-ray data were collected at Southeast Regional Collaborative Access Team (SER-CAT) 22-ID beamline at the Advanced Photon Source, Argonne National Laboratory. SER-CAT is supported by its member institutions, and equipment grants (RR25528, RR028976, and OD027000) from the NIH. Molecular graphics and analyses performed with UCSF ChimeraX Ver. 1.5, developed by the Resource for Biocomputing, Visualization, and Informatics at the University of California,

San Francisco, with support from NIH GM129325 and the Office of Cyber Infrastructure and Computational Biology, National Institute of Allergy and Infectious Diseases.

## Author contributions

S.E.W., R.M.M., P.tD., G.J.I., and A.P.H. designed the study. S.E.W., T.A.S., A.M., C.S., S.P.S., M.vD., K.T.C., M.P.J.W., T.C., J.P., and C.S.H. designed and executed experiments. S.E.W. and T.A.S. crystalized the TGM6-D3:TGFBR2 complex and determined the crystal structure. S.E.W., T.C., and C.S.H. purified recombinant proteins. S.E.W. and A.M. performed ITC measurements. C.S., S.P.S., M.vD. K.T.C., M.P.J.W., T.C., J.P., P.tD., G.J.I., and R.M.M. provided and analyzed signaling activity data. S.E.W and A.P.H wrote the manuscript. P.tD., G.J.I., R.M.M., and A.P.H. provided funding for the study.

## Competing interests

A.H. is a member of the scientific advisory board for TCGFB and is a licensee of intellectual property to Kalivir Immunotherapeutics. The remaining authors declare no competing interests.
