## [Transparent Peer Review file · Nature Communications]

TGM6, a helminth secretory product, mimics TGF- β binding to TGFBR2 to antagonize signaling in fibroblasts

Corresponding Author: Dr Andrew Hinck

Version 0:

Reviewer comments:

Reviewer #1

(Remarks to the Author)

The study by White et al. offers a fascinating look at the intersection of parasitology, structural biology, and immunology, revealing the sophisticated mechanisms employed by the helminth *H. polygyrus* to modulate the TGF- β pathway. Their identification of TGM6 as a cell-type-specific TGF- β antagonist represents a significant breakthrough in our understanding of helminth immune evasion strategies. Unlike its agonist counterpart TGM1, TGM6 inhibits TGF- β signaling in fibroblasts but not T cells due to its divergent D4/D5 domains. This unique property suggests potential applications for developing novel therapeutic agents targeting dysregulated TGF- β signaling in diseases like fibrosis and cancer.

This study demonstrates a high degree of technical rigor and a thorough experimental approach. The combination of structural analysis, binding studies, and cell-based assays provides a compelling and comprehensive exploration of TGM6 function. The authors convincingly show that TGM6 specifically binds T β RII and inhibits TGF- β signaling in fibroblasts, but not T cells. This cell-type specificity is linked to its divergent D4/D5 domains. Additionally, their crystal structure reveals remarkable structural mimicry between the TGM6:T β RII and TGF- β :T β RII interfaces.

The following suggestions could help make the work even more impactful:

- While identification of the TGM6 fibroblast co-receptor would be a major advance, it is likely beyond the scope of this study.
- More clearly outline controls and statistical methods in all experiments, including visual representation of statistical significance in figures.
- Replace "does not signal" with "does not activate TGF- β signaling" for clarity.
- The introduction could be made more engaging by prioritizing key takeaways and emphasizing the broader significance of the findings. Consider reducing some of the specific details about protein domains, amino acid counts, and disulfide bonds.
- Clarify the statement "In TGF- β reporter assays in fibroblasts, TGM6 potently inhibits TGF- β - and TGM1-induced signaling, consistent with its receptor binding profile, including its ability to compete with TGF- β for binding T β RII. However, TGM6 does not inhibit TGF- β or TGM1 signaling in T cells, consistent with its divergent D4 and D5 which have just 42% and 32% identity compared to TGM1 D4 and D5 (Fig. 1A) and its inability to bind CD44." Consider providing a brief explanation of the role of D4/D5 domains after the first sentence. Otherwise, the statement is confusing.

Overall, the manuscript presents original research with potentially wide-reaching implications and is methodologically strong.

Reviewer #2

(Remarks to the Author)

The manuscript by White and colleagues investigated the interesting host-pathogen modulation of TGF β signaling. In this case, a family of proteins, termed TGMs, have multiple domains that engage the different signaling receptors for TGF β signaling. TGM1 has been previously characterized and shown to bind both the type I and type II receptor to bring them into proximity and activate signaling. A targeting or specificity domain is included which binds CD44 and restricts signaling to T cells that express CD44.

Previous work has structurally characterized the T β RII binding domain of TGMs and has indicated a functional and structural

similarity to how T β RII engages TGF β ligands, however the molecular details of these interactions were lacking. This paper provides the first atomic resolution information for how this interaction occurs by solving the structure of TGM6 D3 a TGF β ligand. While a monomeric version of TGF β was used in the analysis, the interactions of D3 with the ligand only occur at the fingertips which are structurally equivalent regardless of whether the ligand is in a dimeric form.

The manuscript provides biochemical details for the interactions using a combination of ITC and cell-signaling assays to dissect the differential affinities of TGM6 D3 with the D3 of TGM1. The conclusion is that the affinity of TGM6 D3 has evolved a higher affinity for T β RII, presumably since it does not have the D1 and D2 domains that are important for Alk5 binding. Thus, in order for it to be an effective inhibitor it needs to replace the binding interactions lost from D1 and D2 to D3 and the D4/5 domains.

Overall, this is a technically sound manuscript that provides new information for how TGM6 inhibits TGF β signaling. An interesting aspect of TGM6 is that inhibition is restricted to fibroblast. The manuscript goes on to show that D45 are required for inhibition and do not bind to CD44. By association with TGM1 which binds CD44 through D4 and D5, the authors make the assumption that a co-receptor provided by the fibroblast is required inhibition. At this point the manuscript transitions into the structural aspects of D3 binding to T β RII leaving the reader with an incomplete picture of D45 and their role in specificity for fibroblast. Could the authors provided evidence, perhaps through FACS analysis, that supports increased binding of TGM6 to fibroblast over other cells? Does just the D45 bind to fibroblasts? Have the authors attempted to identify the co-receptor. If these experiments are unattainable, perhaps the authors can do an analysis of potential difference in co-receptors that are present in the different cell populations. Finally, while these are these interactions specific to murine fibroblast? Given that much of the TGF β signaling molecules are highly conserved with murine vs human it would be interesting to determine if inhibition also occurs with human cells types.

Reviewer #3

(Remarks to the Author)

The NMR analysis carried out in this study is complementary to the other structural aspects of the study and provides key pieces of information regarding the binding of different constructs. In particular, the NMR analysis of TGM6-D3 and T β RII complements the crystal structure determined on this complex, with the significant chemical shift perturbations highlighting the binding between the two proteins. The lack of chemical shift changes for the TGM6 D45 construct is consistent with the conclusions that no binding is occurring. The NMR data appears to be of high quality, carried out using standard procedures suggesting that the overall conclusions drawn have a solid basis. Importantly, the authors have considered the implications of misfolding of the expressed proteins and have shown the data of the individual proteins which are consistent with the proteins being well-structured.

Reviewer #4

(Remarks to the Author)

The study by White, S.E., et al. investigates the activity of TGM6, a peptide secreted by helminth parasite. Unlike TGM1, TGM6 lacks the D1/D2 domains, which are crucial for TGM1's ability to bind to T β R1. However, the D3 domain of TGM6 can bind to T β RII similarly to the D3 domain of TGM1. The authors find that due to TGM6's inability to bind to T β R1, it acts as an antagonist of TGF β signaling. The crystal structure of the TGM6-D3:T β RII complex resembles that of the TGF β 3:T β RII complex. The authors demonstrate that TGM1-dependent inhibition of TGF β signaling is specific to fibroblasts and does not occur in T cells. This study proposes a potential mechanism underlying the cell-type-specific effects of the parasite. The manuscript reveals an activity of TGM6, which has not been studied before, through rigorous biophysical, biochemical, and structural analyses. The experiments are well-executed, and the results are presented clearly. The only major issue of the manuscript is that it does not offer direct evidence for the binding of TGM6 to T β RII, which is essential to be included. The reviewer offers additional recommendations to reinforce the conclusions, as stated below.

Major points:

1. The manuscript fails to offer direct biochemical evidence for the binding of TGM6-D3 to T β RII, aside from its structural resemblance to TGM1-D3. The authors should conduct a biochemical experiment to conclusively demonstrate the interaction between TGM6 and T β RII via the D3 domain.
2. In Figure 6, the authors highlight the similarity between the TGM6-D3:T β RII complex and the T β RII:TGF β 3 complex based on their structures. To validate the amino acid residues crucial for the interface between TGM6-D3 and T β RII, the authors should introduce mutations at for example Arg95, Tyr80, or Arg82 in TGM6 and demonstrate that these mutations impair TGM6's ability to inhibit TGF β 1- or TGM1-induced signaling.
3. The authors state that D3 domain alone is not sufficient to inhibit TGF β signaling. The authors should generate D3-D4 or D3-D5 and examine whether the addition of D4 or D5 to D3 is sufficient to restore the inhibitory activity of TGM6 in fibroblasts.
4. Is the chimera of TGM6-D3-TGM1-D4D5 allows the inhibition of TGF β signaling in both fibroblasts and T cells?
5. In p.10, the authors state that TGM6-D45 does not bind CD44. The result should be included in the manuscript.
6. As a readout of the TGF β signaling, the authors should include the transcriptional activity of Smad2/3, such as SBE-Luc activity in addition to phosphorylation of Smad2/3.
7. In the figure legend of Figure 6, although the structure is TGF β 3:T β RII, the legend states "TGF β 1 and T β RII are shaded burnt orange..."

Version 1:

Reviewer comments:

Reviewer #1

(Remarks to the Author)

White et al. have effectively addressed the reviewers' comments by clearly describing their experimental controls and statistical methods, using precise scientific language, and clarifying how the D4/D5 domains of TGM6 allow it to bind to fibroblasts and epithelial cells selectively. The revised manuscript provides a more engaging and accessible read while maintaining scientific rigor. Through the combined use of structural insights, binding studies, and cell-based assays, the authors present a comprehensive picture of the role of TGM6 in helminth immune evasion.

Reviewer #2

(Remarks to the Author)

The authors have addressed the suggestions from the review process. It is acceptable to leave information about the discovery of a co-receptor to another manuscript.

Reviewer #4

(Remarks to the Author)

The authors did not address all the points raised by this reviewer. Nevertheless, the manuscript is well-executed overall and presents interesting findings that justify its publication.

Reviewer Comments are in Grey and Author responses are in Blue.

Reviewer #1 (Remarks to the Author):

The study by White et al. offers a fascinating look at the intersection of parasitology, structural biology, and immunology, revealing the sophisticated mechanisms employed by the helminth *H. polygyrus* to modulate the TGF- β pathway. Their identification of TGM6 as a cell-type-specific TGF- β antagonist represents a significant breakthrough in our understanding of helminth immune evasion strategies. Unlike its agonist counterpart TGM1, TGM6 inhibits TGF- β signaling in fibroblasts but not T cells due to its divergent D4/D5 domains. This unique property suggests potential applications for developing novel therapeutic agents targeting dysregulated TGF- β signaling in diseases like fibrosis and cancer.

This study demonstrates a high degree of technical rigor and a thorough experimental approach. The combination of structural analysis, binding studies, and cell-based assays provides a compelling and comprehensive exploration of TGM6 function. The authors convincingly show that TGM6 specifically binds T β RII and inhibits TGF- β signaling in fibroblasts, but not T cells. This cell-type specificity is linked to its divergent D4/D5 domains. Additionally, their crystal structure reveals remarkable structural mimicry between the TGM6:T β RII and TGF- β :T β RII interfaces.

:

Overall, the manuscript presents original research with potentially wide-reaching implications and is methodologically strong.

We thank the Reviewer for these broadly positive remarks pertaining to both the breadth and impact, but also the technical quality, of our study.

The following suggestions could help make the work even more impactful:

We are grateful for the suggestions and have modified the manuscript to take these into account, as described in further detail below.

- While identification of the TGM6 fibroblast co-receptor would be a major advance, it is likely beyond the scope of this study.

We have made some progress identifying the co-receptor(s) responsible for the cell type specific inhibition by TGM6, however more work is required to fully understand how this (these) co-receptor(s) function. We therefore appreciate the recognition by the Reviewer that this represents an entire new study to extend current understanding and therefore is not practical (or wise) to include as part of this study.

- More clearly outline controls and statistical methods in all experiments, including visual representation of statistical significance in figures.

We agree that we did not present the controls and experimental replicas and repeats as thoroughly as we should have. We have carefully gone through the methods, figure legends, and figures and now present the controls and experimental replicas and repeats for *all* experiments, both visually and in the text and legends:

- ITC experiments are shown as an overlay of 2 – 3 experimental repeats, with errors bars indicating the standard deviation of the integrated heats at each point in the titration; also we show the fit residuals for fitting of the integrated heats to a standard binding isotherm (Fig. 2A, C, Fig. 3, Fig. 8A,B, Fig. 5D,E, and S9).
- Replicate data points and standard deviations are shown for *all* TGF- β reporter data (Fig. 1B, 4, 5A-C, Fig. 8C, D, F, G, Fig. S7, and Fig. S12A-B); details of the number of technical and experimental repeats and controls are provided in the figure legends.

• Replace "does not signal" with "does not activate TGF- β signaling" for clarity.

We have made this change (Line 100).

• The introduction could be made more engaging by prioritizing key takeaways and emphasizing the broader significance of the findings. Consider reducing some of the specific details about protein domains, amino acid counts, and disulfide bonds.

We have removed the details about domain amino acid counts and have bolstered the key takeaways in the concluding paragraph (see Lines 124-129) – we nonetheless feel it is important to discuss the domain structure of TGM1 and how this differs in TGM6, as this provides essential background important for understanding why TGM1 is an agonist, but TGM6 is an antagonist.

• Clarify the statement “In TGF- β reporter assays in fibroblasts, TGM6 potently inhibits TGF- β - and TGM1-induced signaling, consistent with its receptor binding profile, including its ability to compete with TGF- β for binding T β RII. However, TGM6 does not inhibit TGF- β or TGM1 signaling in T cells, consistent with its divergent D4 and D5 which have just 42% and 32% identity compared to TGM1 D4 and D5 (Fig. 1A) and its inability to bind CD44.” Consider providing a brief explanation of the role of D4/D5 domains after the first sentence. Otherwise, the statement is confusing.

We have modified the text as suggested (Lines 106-118), thank you for this helpful suggestion.

Reviewer #2 (Remarks to the Author):

The manuscript by White and colleagues investigated the interesting host-pathogen modulation of TGF β signaling. In this case, a family of proteins, termed TGMs, have multiple domains that engage the different signaling receptors for TGF β signaling. TGM1 has been previously characterized and shown to bind both the type I and type II receptor to bring them into proximity and activate signaling. A targeting or specificity domain is included which binds CD44 and restricts signaling to Tcells that express CD44.

Previous work has structural characterized the T β RII binding domain of TGMs and has indicated a functional and structural similarity to how T β RII engages TGF β ligands, however the molecular details of these interactions were lacking. This paper provides the first atomic resolution information for how this interaction occurs by solving the structure of TGM6 D3 a TGF β ligand. While a monomeric version of TGF β was used in the analysis, the interactions of D3 with the ligand only occur at the fingertips which are structurally equivalent regardless of whether the ligand is in a dimeric form.

The manuscript provides biochemical details for the interactions using a combination of ITC and cell-signaling assays to dissect the differential affinities of TGM6 D3 with the D3 of TGM1. The conclusion is that the affinity of TGM6 D3 has evolved a higher affinity for T β RII, presumably since it does not have the D1 and D2 domains that are important for Alk5 binding. Thus, in order

for it to be an effective inhibitor it needs to replace the binding interactions lost from D1 and D2 to D3 and the D4/5 domains.

Overall, this is a technically sound manuscript that provides new information for how TGM6 inhibits TGF β signaling.

We thank the Reviewer for these broadly positive remarks regarding both the impact and technical quality of our study.

An interesting aspect of TGM6 is that inhibition is restricted to fibroblast. The manuscript goes on to show that D45 are required for inhibition and do not bind to CD44. By association with TGM1 which binds CD44 through D4 and D5, the authors make the assumption that a co-receptor provided by the fibroblast is required inhibition. At this point the manuscript transitions into the structural aspects of D3 binding to T β RII leaving the reader with an incomplete picture of D45 and their role in specificity for fibroblast. Could the authors provided evidence, perhaps thorough FACS analysis, that supports increased binding of TGM6 to fibroblast over other cells? Does just the D45 bind to fibroblasts? Have the authors attempted to identify the co-receptor. If these experiments are unattainable, perhaps the authors can do an analysis of potential difference in co-receptors that are present in the different cell populations.

We appreciate the Reviewer's comments and as explained in response to Reviewer 1, while we are working to identify and characterize the co-receptor(s) that enable the function of TGM6, this represents an entire new study in itself and is not yet complete (and hence we are reticent about presenting an abbreviated and incomplete study that might raise more questions than it answers). While we present what we believe to be strong data showing that the co-receptor binding domains of TGM6 (D45) are required for its inhibitory activity (**Fig. 5A-C**) and that the co-receptor binding specificity of TGM6 is different from that of TGM1 (**Fig.5D-F**), we nonetheless acknowledge that our data showing the range of cells on which TGM6 acts and how this is different from that of the agonist TGM1 is limited.

While not explicitly requested by the reviewer, we have therefore investigated two additional mouse cell lines, EL4, a type of T-cell and NM18, a subclone of NMuMG breast epithelial cells, to better understand the range of cells in which TGM6 inhibits TGF- β signaling. We found that TGM6 did not inhibit signaling in EL4 cells, while it does inhibit signaling in NM18 epithelial cells. We have included this data as part of the revised manuscript (**Fig. S6**) and append below for easy reference. In accord with these new findings, we reassert our claim that TGM6 potently inhibits TGF- β signaling in fibroblasts but not T-cells, but clarify that TGM6 inhibition is not restricted only to fibroblasts (see Lines 51 and 55 in Abstract, Line 104 in Introduction, Lines 243-246, 256 in Results, and Lines 429-433 and 443-445 in Discussion).

While we have not yet used FACS to correlate cell binding with TGM6 inhibitory activity, we have used FACS to determine whether TGM6-D45 is sufficient to enable binding to fibroblasts, as suggested by the reviewer, and we find that it is (see **Fig. 1 for Response** inserted below). Since we would like to present this data as part of our future manuscript detailing the TGM6 coreceptor(s) and how they function, we have chosen to present this data below as part of this response rather than as part of the revised manuscript.

Lastly, to investigate the extent to which the inhibitory activity of TGM6 is dependent on its D45 co-receptor binding domains, we generated conditioned medium containing either TGM6 or a chimeric TGM6, TGM6-D3-TGM1-D45, with D3 from TGM6 and D45 from TGM1 (both with a myc tag appended to the N-terminus). We validated the presence of TGM6 and the TGM6-D3-TGM1-D45 chimera in the conditioned medium by Western blotting with an anti-myc antibody and we assayed the inhibitory activity of comparable quantities of protein in either wild type NM18 cells with CD44 present or with CD44 knocked out (KO). We found that TGM6 and TGM6-D3-TGM1-D45 conditioned medium, but not also the TGM6-D3-TGM1-D45 conditioned medium, inhibited TGF- β reporter activity in the NM18 cells in which CD44 had been knocked out. This shows that the inhibitory

activity of TGM6 D3 is highly dependent upon the identity of its co-receptor (D45) binding domain, with inhibition only occurring in cells in which the cognate co-receptor is present. Since we would like to present this data as part of our manuscript detailing the TGM6 coreceptor(s) and how they function, we have chosen to present this data below as part of this response (**Fig. 2 for-Response**) rather than as part of the revised manuscript.

Overall, we believe that the data already included in the paper showing that inhibition by TGM6 is dependent on D45 (**Fig. 5A-C**) and that these domains do not bind CD44 (**Fig. 5D-F**), together with this new data, provides strong support for our claim that TGM6 D45 is critical for cell-specific antagonism of TGF- β signaling by TGM6. While we have chosen to include only some of this data in the current manuscript, we plan to include the data that we do not report here in a future manuscript that describes the TGM6 co-receptor(s) and the mechanism by which they function.

Finally, while these are these interactions specific to murine fibroblast? Given that much of the TGF β signaling molecules are highly conserved with murine vs human it would be interesting to determine if inhibition also occurs with human cells types.

We analyzed inhibition of TGF- β signaling in cultured human fibroblasts by TGM6 and found that in contrast to the potent inhibition of TGF- β signaling in murine fibroblasts, there is no detectable inhibition by conditioned medium containing TGM6 or the TGM6-D3-TGM1-D45 chimera (**Fig-for-Response-3**).

In light of the high conservation of the ectodomain of mouse and human T β RII with just one conservative substitution among all interface residues (Glu142 in place of Asp142), we suspect that the difference is related to species differences of the co-receptor(s), including possible differences in the splice variants that predominate on the cell surface. Our supposition is based on our previous finding that the agonist TGM1 was largely ineffective in activating TGF- β signaling in most human cell lines, even though CD44 was present, and that species differences between mouse and human CD44 (and the different splice forms that are present) were largely responsible [van Dinther, et al (2023 *Proc. Natl. Acad. Sci.*, 120, e2302370120)]. Hence, there is precedence for this mechanism, which we are now investigating and will report in our upcoming manuscript focused on identification of the TGM6 coreceptor(s) and the mechanism by which they function.

Reviewer #3 (Remarks to the Author):

The NMR analysis carried out in this study is complementary to the other structural aspects of the study and provides key pieces of information regarding the binding of different constructs. In particular, the NMR analysis of TGM6-D3 and T β RII complements the crystal structure determined on this complex, with the significant chemical shift perturbations highlighting the binding between the two proteins. The lack of chemical shift changes for the TGM6 D45 construct is consistent with the conclusions that no binding is occurring. The NMR data appears to be of high quality, carried out using standard procedures suggesting that the overall conclusions drawn have a solid basis. Importantly, the authors have considered the implications of misfolding of the expressed proteins and have shown the data of the individual proteins which are consistent with the proteins being well-structured.

We thank the Reviewer for these positive remarks regarding the technical quality of our study.

Reviewer #4 (Remarks to the Author):

The study by White, S.E., et al. investigates the activity of TGM6, a peptide secreted by helminth parasite. Unlike TGM1, TGM6 lacks the D1/D2 domains, which are crucial for TGM1's ability to bind to TbRI. However, the D3 domain of TGM6 can bind to TbRII similarly to the D3 domain of TGM1. The authors find that due to TGM6's inability to bind to TbRI, it acts as an antagonist of TGF β signaling. The crystal structure of the TGM6-D3:TbRII complex resembles that of the TGF β 3:TbRII complex. The authors demonstrate that TGM1-dependent inhibition of TGF β signaling is specific to fibroblasts and does not occur in T cells. This study proposes a potential mechanism underlying the cell-type-specific effects of the parasite. The manuscript reveals an activity of TGM6, which has not been studied before, through rigorous biophysical, biochemical, and structural analyses. The experiments are well-executed, and the results are presented clearly.

We thank the Reviewer for these broadly positive remarks regarding both the impact and technical quality of our study.

The only major issue of the manuscript is that it does not offer direct evidence for the binding of TGM6 to TbRII, which is essential to be included. The reviewer offers additional recommendations to reinforce the conclusions, as stated below.

We do not understand this criticism as we provide:

1. ITC and NMR binding data for this interaction (**Fig. 2**).

2. Clear binding isotherm in the ITC data that we can readily fit to 1:1 binding model for both full-length TGM6 and TGM6-D3 (**Fig. 2A, C, Table S1**)
3. Large chemical shift changes are observed for some, but not all of the residues in the NMR spectrum of ¹⁵N-TGM6-D3 upon addition of unlabeled TβR11, indicating specific binding (**Fig. 2B**).
4. Crystal structure of TGM6-D3 bound to TβR11 determined to a resolution of 1.4 Å (**Figs. 6 and 7**).
5. Crystal structure shows that TGM6-D3 binds TβR11 through an interface that is fully consistent with expectations based on previous interface mapping with TGM1-D3 and TβR11 [Mukundan, et al (2022) *J. Biol. Chem*, 298, 101994].

We further note that the interface observed in the crystal structure is consistent with a) ITC competition binding data with TGF-β shown in **Fig. 3**, an extensive analysis of TGM6-D3 and TβR11 mutants reported in **Table S6**, and functional studies of TGM6-D3 and TGM1-D3 mutants in signaling assays reported in **Fig. 7**. We therefore believe we have provided ample evidence demonstrating that TGM6-D3 directly binds TβR11.

Major points:

1. The manuscript fails to offer direct biochemical evidence for the binding of TGM6-D3 to TβR11, aside from its structural resemblance to TGM1-D3. The authors should conduct a biochemical experiment to conclusively demonstrate the interaction between TGM6 and TβR11 via the D3 domain.

We respectfully disagree with this comment for the reasons mentioned in the two paragraphs above. We would further assert that not only have we already provided ample biochemical data demonstrating direct binding of TGM6-D3 to TβR11, but we have provided the strongest possible data, namely a very high resolution crystal structure validated by an extensive mutational analysis (see the data reported in **Table S6** and the accompanying description of the results on lines 304-348 and 350-406) of the revised manuscript; see also the response to point number 2 below).

2. In Figure 6, the authors highlight the similarity between the TGM6-D3:TβR11 complex and the TβR11:TGFβ3 complex based on their structures. To validate the amino acid residues crucial for the interface between TGM6-D3 and TβR11, the authors should introduce mutations at for example Arg95, Tyr80, or Arg82 in TGM6 and demonstrate that these mutations impair TGM6's ability to inhibit TGFβ1- or TGM1-induced signaling.

While we have not analyzed the effects of mutating these three residues in signaling assays, we have mutated these and studied binding of the purified mutant proteins (R95A, Y80A, Y80F, R82A, and R82S) to purified TβR11 using ITC. We found that all of the mutant proteins significantly perturbed binding, as summarized below (and as reported in **Table S6**).

We note that the Y80A mutant not only had the largest perturbation of binding among this subset of mutants, it also had the largest perturbation of binding (134-fold increase in K_D) among all TGM6 mutants tested (**Table S6**). While not as large, the R82A and R95A mutants each perturbed binding by more than 10-fold (11.8- and 10.6-fold increased in K_D , respectively), which are still quite large perturbations for single-residue substitutions. We believe that this data, together with our demonstration that the most strongly perturbed of these mutants, such as Y80A, did not perturb the overall folding of the protein (**Fig. S10**), provides strong evidence for the importance of these residues for binding as suspected based on the crystal structure.

We note that while we have not analyzed individual mutants in signaling assays, we did perform an in-depth analysis to identify residues responsible for the roughly 5-fold greater binding affinity of TGM6-D3 for T β RII compared to TGM-D1. We found that the QRRG tetrapeptide of TGM6, which includes Arg82 at the second position, is responsible for both the higher binding affinity of TGM6-D3 for T β RII compared to TGM1-D3 (which includes a corresponding KSGT tetrapeptide) and is required for potent antagonistic potency of TGM6 (**Table S6 and Fig. 7**).

In summary, while it would also be interesting to evaluate single mutants such as R95A, Y80A, Y80F, R82A, and R82S mutants in signaling assays (as we have already successfully done for the TGM6-D3/TGM1-D3 QRRG/KSGT tetrapeptide swaps), we believe that our evaluation of these, and other mutants in binding assays provides ample evidence supporting the importance of the interface residues that we observe in the crystal structure.

3. The authors state that D3 domain alone is not sufficient to inhibit TGF β signaling. The authors should generate D3-D4 or D3-D5 and examine whether the addition of D4 or D5 to D3 is sufficient to restore the inhibitory activity of TGM6 in fibroblasts.

We have not tested these constructs, and while we think doing so is important for understanding how TGM6 interacts with its coreceptor(s), it is not required to support the claims of our paper (that binding of co-receptors by TGM6 is essential for its inhibitory activity and that its co-receptor specificity is different than that of TGM1). We therefore view this as part of a future study focused on identifying and characterizing the co-receptor(s) for TGM6.

4. Is the chimera of TGM6-D3-TGM1-D4D5 allows the inhibition of TGF β signaling in both fibroblasts and T cells?

We have not yet tested the TGM6-D3-TGM1-D4D5 chimera for inhibition of TGF- β /TGM signaling in either mouse fibroblasts or T-cells. However, as noted above in our response to Reviewer #2, we did test the ability of the TGM6-D3-TGM1-D4D5 chimera to inhibit TGF- β signaling in NM18 cells in which CD44 had and had not been knocked out (see **Fig. 2 for-Response** above), and found that the chimera could potentially inhibit signalling when CD44 was present, but not when it was knocked out. While not T-cells, we believe this data in NM18 cells would be reflective of that for T-cells as we have shown before that the function of TGM1 in T-cells is dependent on the presence of CD44 [van Dinther, et al (2023 *Proc. Natl. Acad. Sci.*, 120, e2302370120)]. We do know that TGM1 is active in mouse fibroblasts and that this dependent on its CD44-binding D45 and the presence of CD44, which is expressed at intermediate levels in these cells; hence we would the TGM6-D3-TGM1-D4D5 chimera to also be active in mouse fibroblasts is present, but not when it is knocked out.

5. In p.10, the authors state that TGM6-D45 does not bind CD44. The result should be included in the manuscript

We agree and we note this data was already included in the manuscript (**Fig. 5F**, with **Fig. 5D-E** serving as positive control for the mCD44 that was used for these measurements). We do not have a comparable positive control for TGM6-D45 since we have not definitively identified its co-receptor, though we definitively show in **Fig. S1** that the TGM6-D45 that we isolated and used for these experiments is natively folded. We are therefore confident that TGM6 does not bind CD44, as claimed.

6. As a readout of the TGF β signaling, the authors should include the transcriptional activity of Smad2/3, such as SBE-Luc activity in addition to phosphorylation of Smad2/3.

We provide TGF- β /TGM-induced Smad 2/3 phosphorylation data to assess the inhibitory activity of TGM6 and variants in **Fig. 7E, H, I** and **Fig. S11C**. In **Figures 7** and **S11**, we note we also already provide cell-based SBE-based data, included as **Fig. 7D, F, G** and **Fig. S11A, B** to complement the Smad 2/3 phosphorylation data in **Fig. 7E, H, I** and **Fig. S11C**, respectively. We note that the MFB-F11 reporter we are using has the same SBE as traditional TGF- β luciferase reporter cell lines, the only difference being that the readout is secreted alkaline phosphatase instead of cytoplasmic luciferase [Tesseur, et. al (2006) BMC Cell Biol. 7, 15]. We are therefore confident, based on this complementary assay data, about our conclusions regarding the signaling activity of the TGM6/1 QRRG and KSGT chimeras that we report.

In the figure legend of Figure 6, although the structure is TGF β 3:TbRII, the legend states “TGF β 1 and TbRII are shaded burnt orange...”

We thank the reviewer for bringing this error to our attention; this has been corrected.

Other Revision Notes

While not requested, we further refined the structure of the TGM6-D3:T β RII complex. While this did not change any of the conclusions previously drawn from the structure, the data statistics have improved, with 2 - 3% improvements in the R-factors (R_{work} and R_{free}) relative to the previous refinement. We have updated the text, the X-ray data table (**Table S5**), and our PDB deposition (8GDT) accordingly.